# CERTAIN: Context Uncertainty-aware One-Shot Adaptation for Context-based Offline Meta Reinforcement Learning

**Hongtu Zhou** [* 1]  **Ruiling Yang** [* 1]  **Yakun Zhu** [1]  **Haoqi Zhao** [1]  **Hai Zhang** [1]  **Di Zhang** [1]  **Junqiao Zhao** [1]
**Chen Ye** [1]  **Changjun Jiang** [1]

## Abstract

Existing context-based offline meta-reinforcement learning (COMRL) methods primarily focus on task representation learning and given-context adaptation performance. They often assume that the adaptation context is collected using task-specific behavior policies or through multiple rounds of collection. However, in real applications, the context should be collected by a policy in a one-shot manner to ensure efficiency and safety. We find that inherent context ambiguity across multiple tasks and out-of-distribution (OOD) issues due to distribution shift significantly affect the performance of one-shot adaptation, which has been largely overlooked in most COMRL research. To address this problem, we propose using heteroscedastic uncertainty in representation learning to identify ambiguous and OOD contexts, and train an uncertainty-aware context collecting policy for effective one-shot online adaptation. The proposed method can be integrated into various COMRL frameworks, including classifier-based, reconstruction-based and contrastive learning-based approaches. Empirical evaluations on benchmark tasks show that our method can improve one-shot adaptation performance by up to 36% and zero-shot adaptation performance by up to 34% compared to existing baseline COMRL methods.

## 1. Introduction

Reinforcement learning (RL) has achieved remarkable success in a wide range of domains, including game play-

ing(Mnih, 2013; Lample & Chaplot, 2017), robotic control(Nguyen & La, 2019), and recommendation systems (Zheng et al., 2018). However, RL faces a critical challenge of low sample efficiency due to its reliance on extensive interactions with the environment, a limitation that becomes even more pronounced in multi-task settings (Levine et al., 2020; Li et al., 2020a).

Context-based offline meta reinforcement learning (COMRL) provides a promising solution to improve sample efficiency in multi-task environments by leveraging offline datasets for training and enabling fast adaptation to new tasks with minimal online samples (Li et al., 2020b). COMRL treats task-specific trajectories or transitions as context information and operates in two key phases: the offline meta-training and the online adaptation. In the first phase, a context encoder is learned through task representation learning, alongside a meta-policy conditioned on the learned task representations. During the second phase, the meta-policy adapts to new tasks by conditioning on the task representation inferred from the online-collected context.

Despite its promise, COMRL encounters two major challenges, as depicted in Figure 1 left: context ambiguity and out-of-distribution (OOD). Context ambiguity arises when a context fails to uniquely infer a task because it appears across multiple tasks (Dorfman et al., 2021). Context OOD, on the other hand, occurs when a context falls outside the task distribution of the offline dataset. Both issues can lead the context encoder to infer unreliable task representations, resulting in suboptimal policy performance.

Most existing COMRL methods predominantly focus on task representation learning (Li et al., 2020b; Gao et al., 2024; Li et al., 2024). These methods often assume that contexts are either in-distribution, thereby avoiding OOD issues, or can be collected through multiple rounds, which mitigates the impact of ambiguity. While CORRO (Yuan & Lu, 2022) attempts to address OOD challenges, its applicability is limited to $(s, a)$ pairs observed in the dataset, and it fails to generalize to truly OOD $(s, a)$ scenarios. However, in many practical scenarios, context collection is constrained to a one-shot manner due to safety and efficiency considera-

---

*Equal contribution [1]School of Computer Science and Technology, Tongji University, Shanghai, China. Correspondence to: Junqiao Zhao <zhaojunqiao@tongji.edu.cn>.

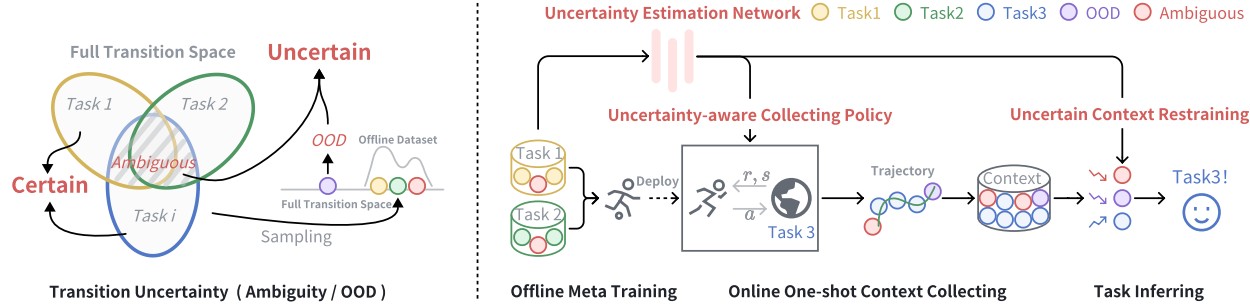

*Figure 1.* Left: Contexts consist of certain task-specific transitions and uncertain transitions that are either ambiguous or OOD. Right: Our approach involves training an uncertainty estimation network and an uncertainty-aware context collecting policy to restrain the detrimental influence of uncertain contexts on task inference during online adaptation.

tions, making context ambiguity and OOD issues even more challenging.

To address these challenges, we propose CERTAIN: Context Uncertainty-aware One-Shot Adaptation. The core insight behind CERTAIN is that both ambiguous and OOD contexts contribute to high task representation uncertainty, whereas contexts that reliably distinguish tasks exhibit lower uncertainty (Figure 1, left).

Building on this observation, we introduce an uncertainty estimation network during the offline meta-training phase to estimate context uncertainty alongside task representation learning (Figure 1, right). By leveraging these uncertainty estimates, CERTAIN effectively identifies ambiguous and OOD contexts, mitigating their negative impact on task inference during online adaptation. To enable fast and accurate task inference, we further propose training an uncertainty-aware context collecting policy that prioritizes the collection of low-uncertainty contexts during the online adaptation phase.

CERTAIN is designed as a plug-in framework that seamlessly integrates into existing COMRL methods, including classifier-based, reconstruction-based, and contrastive learning-based approaches, to enhance robustness against context ambiguity and OOD issues during online adaptation. We validate CERTAIN through extensive experiments on a range of benchmark tasks. The results show that CERTAIN significantly improves the performance of existing COMRL methods in the one-shot adaptation setting, achieving up to a 36% increase in average return. Furthermore, in the zero-shot adaptation setting—where the meta-policy is directly applied to new tasks without any online adaptation—CERTAIN delivers up to a 34% improvement in average return, outperforming state-of-the-art (SoTA) COMRL methods.

## 2. Related Work

### 2.1. Offline Reinforcement Learning

Offline reinforcement learning (Offline RL) aims to learn a policy from a static dataset without requiring interactions with the environment, significantly improving sample efficiency (Levine et al., 2020). Recent research has focused on addressing key challenges such as value overestimation (Kumar et al., 2020; Li et al., 2022; Singh et al., 2023; Mao et al., 2024) and distributional shift (Fujimoto et al., 2019; Kumar et al., 2019; Kostrikov et al., 2021; Brandfonbrener et al., 2021; Ran et al., 2023; Zhang & Tan, 2024). Our work adheres to the offline RL setting, targeting the learning of the task representation and the meta-policy from multi-task offline datasets.

### 2.2. Meta Reinforcement Learning

Meta reinforcement learning (Meta RL) seeks to train a meta-policy that generalizes across multiple tasks, enabling rapid adaptation to new tasks. Context-based methods (Duan et al., 2016; Rakelly et al., 2019; Fakoor et al., 2019; Zintgraf et al., 2019; Humplik et al., 2019; Wang et al., 2023b; Chu et al., 2024) frame Meta RL as a partially observable Markov decision process (POMDP), where historical trajectories serve as the context. These methods extract task-specific information from the context and treat it as a hidden state to guide decision-making. Gradient-based methods (Finn et al., 2017; Finn & Levine, 2017; Xu et al., 2018; Rothfuss et al., 2018; Xu et al., 2020; Liu et al., 2022; Xu & Zhu, 2024) aim to learn a meta-policy that provides a strong initialization for task-specific policies. Adaptation to new tasks is achieved by fine-tuning the meta-policy using meta-gradients computed from online few-shot samples. Our work builds on the context-based Meta RL framework, which is better suited for real-world applications due to its practicality and scalability.

## 2.3. Offline Meta Reinforcement Learning

Offline meta reinforcement learning (OMRL) focuses on learning a meta-policy from multi-task offline datasets to enable rapid adaptation to new tasks (Li et al., 2020a). Similar to Meta RL, OMRL methods are categorized into gradient-based (Mitchell et al., 2021) and context-based (Li et al., 2020b) approaches. The key distinction is that OMRL operates under an offline RL setting.

Our work is grounded in context-based OMRL methods. Recent COMRL approaches (Li et al., 2020b; Yuan & Lu, 2022; Gao et al., 2024) emphasize task representation learning during the meta-training phase, which can be unified under the perspective of mutual information maximization (Li et al., 2024). Other works (Gao et al., 2024; Wang et al., 2023a) address the challenge of context collection during the online adaptation phase. Our method aligns with the latter category, focusing on improving context collection for robust one-shot adaptation.

## 2.4. Uncertainty Estimation

Uncertainty can be categorized into epistemic uncertainty and aleatoric uncertainty. Prior work has tackled epistemic uncertainty estimation using ensemble methods (Lakshminarayanan et al., 2017) and Bayesian neural networks (Gal & Ghahramani, 2016), while aleatoric uncertainty is typically modeled via heteroscedasticity by parameterizing the variance of the output distribution (Kendall & Gal, 2017). However, the heteroscedastic uncertainty obtained with the above method can also be affected by the model's capacity and the training dynamics. Building upon the heteroscedastic uncertainty framework, we propose an approach to estimate context uncertainty in the setting of COMRL.

## 3. Problem Formulation

### 3.1. Preliminaries

A task in RL is formally represented as a Markov Decision Process (MDP) $\mathcal{M} = \langle \mathcal{S}, \mathcal{A}, \mathcal{P}, \mathcal{R} \rangle$, where $\mathcal{S}$ denotes the state space, $\mathcal{A}$ the action space, $\mathcal{P}$ the transition dynamics, and $\mathcal{R}$ the reward function. In COMRL, each task is assumed to be sampled from a task distribution:

$$\mathcal{M}_i = \langle \mathcal{S}, \mathcal{A}, \mathcal{P}_i, \mathcal{R}_i \rangle \sim p(\mathcal{M}) \qquad (1)$$

The context $c$ for task $\mathcal{M}_i$ is defined as a set of transition tuples:

$$c = \{(s_j, a_j, r_j, s'_j)\}_{j=1}^K \qquad (2)$$

Given an offline dataset $\mathcal{D}$ containing $M$ tasks:

$$\mathcal{D} = \{\mathcal{D}_i = \{(s_j, a_j, r_j, s'_j)\}_{j=1}^N\}_{i=1}^M \qquad (3)$$

the objective of offline meta-training in COMRL is to learn a context encoder $q_\phi(z|c)$ and a meta-policy $\pi_\theta(a|s, z)$ that

maximize the expected return $J$ across tasks:

$$\max_{\phi, \theta} \mathbb{E}_{\mathcal{M}_i \sim p(\mathcal{M})} \left[ \mathbb{E}_{c \sim \mathcal{D}_i, z \sim q_\phi(z|c)} J(\pi_\theta(a|s, z)) \right] \qquad (4)$$

here, $z$ is the latent task representation inferred from the context $c$, specifically:

$$z = \frac{1}{K} \sum_{j=1}^K q_\phi(z_j | s_j, a_j, r_j, s'_j) \qquad (5)$$

In the online adaptation phase, trajectories are collected for the unseen task $\mathcal{M}'$ using a context collecting policy $\pi_c$ to form the adaptation context $c'$:

$$c' = \{(s_j, a_j, r_j, s'_j)\}_{j=0}^T \sim \langle \mathcal{M}', \pi_c \rangle \qquad (6)$$

Subsequently, the adapted policy $\pi_\theta(a|s, z')$ is derived based on the inferred task representation $z' = q_\phi(z|c')$.

### 3.2. Context Uncertainty in Task Inference

In COMRL, the task representation $z$ is inferred from the context $c$ using a context encoder $q_\phi(z|c)$. Assuming a reasonable $q_\phi$, the reliability of task inference based on $z$ heavily depends on the context $c$. Two major factors that lead to unreliable task inference are context ambiguity and OOD contexts. Before formally defining ambiguity and OOD, we first define the probability of a context $c = \{(s_j, a_j, r_j, s'j)\}_{j=1}^K$ given a task $\mathcal{M}$ and a policy $\pi$ as:

$$p(c|\mathcal{M}, \pi) = \prod_{j=1}^K \rho(s_j) \pi(a_j|s_j) p_{\mathcal{M}}(r_j, s'_j | s_j, a_j) \qquad (7)$$

where $\rho(s)$ denotes the initial state distribution, and $p_{\mathcal{M}}(s', r|s, a)$ is the model of the task $\mathcal{M}$.

With this formulation, we define context ambiguity and OOD contexts as follows:

**Definition 3.1. Context Ambiguity:** Context ambiguity occurs when a given context $c$ corresponds to at least two distinct tasks $\mathcal{M}_i$ and $\mathcal{M}_j$, such that $p(c|\mathcal{M}_i) > 0$ and $p(c|\mathcal{M}_j) > 0$, where $p(c|\mathcal{M}) := \int p(c|\mathcal{M}, \pi) d\pi$.

**Definition 3.2. OOD Context:** A context $c$ is considered OOD if it cannot be sampled by the behavior policy $\pi_\beta$ within the offline dataset task $\mathcal{M}$, i.e., $p(c|\mathcal{M}, \pi_\beta) = 0$.

**Definition 3.3. Context Uncertainty:** Contexts affected by either ambiguity or OOD issues are likely to produce unreliable task representations for task inference, which we refer to as **Uncertain contexts**. Conversely, a context that is neither ambiguous nor OOD is defined as a **Certain context**, capable of reliably inferring a task based on $z$. To quantify the reliability of a context for task inference, we define **Context Uncertainty** as $\sigma$, which captures the degree of uncertainty in task inference for a given context.

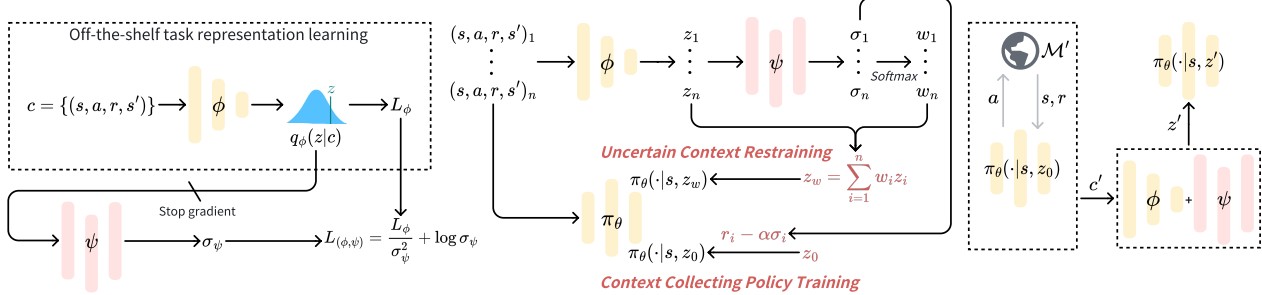

(a) Uncertainty-aware representation learning     (b) Uncertainty-aware meta-policy learning     (c) Uncertainty-aware adaptation

*Figure 2.* Overview of the proposed method: (a) combining off-the-shelf task representation learning with heteroscedastic uncertainty framework, (b) integrating Uncertainty in meta-policy learning and context collecting policy learning, and (c) uncertainty-aware one-shot adaptation in unseen task.

### 3.3. One-shot Adaptation Challenges

In the online adaptation phase, most existing COMRL methods assume that the adaptation context $c'$ is collected by the task-specific behavior policy $\pi_\beta$ for the unseen task $\mathcal{M}'$, following a distribution similar to the offline dataset $\mathcal{D}$. Alternatively, these methods assume that $c'$ can be collected over multiple rounds using an exploratory policy, e.g. a random policy, to obtain more certain contexts, thereby mitigating the impact of uncertain contexts. However, under the one-shot adaptation setting, the adaptation context $c'$ is derived from a single trajectory collected by the context collecting policy $\pi_c$. This restriction substantially increases the likelihood of encountering uncertain contexts, which can impair task inference based on the task representation, thereby degrading policy performance.

## 4. Method

We now present our method, CERTAIN, which consists of three key components: (a) uncertainty-aware task representation learning, (b) uncertainty-aware meta-policy learning, and (c) uncertainty-aware adaptation. Figure 2 provides an overview of our method.

### 4.1. Uncertainty-aware Task Representation Learning

Building on the heteroscedastic uncertainty framework proposed in (Kendall & Gal, 2017), we introduce a method to evaluate context uncertainty in task representation learning by slightly modifying the heteroscedastic loss. This formulation allows the context uncertainty to be seamlessly integrated into the three primary paradigms of task representation learning in COMRL: (1) classifier-based methods, (2) reconstruction-based methods, and (3) contrastive learning-based methods. All these methods share the context encoder $q_\phi(z|c)$ as a core component but differ in their training objectives.

jectives.

Classifier-based methods (Zhang et al., 2025) train the context encoder by predicting the task label $y$ using a classifier $p_\varphi(y|z)$. The objective is to minimize the cross-entropy loss between the predicted task label and the ground truth:

$$\mathcal{L}_{\text{cls}} = -\mathbb{E}_{(s,a,r,s')\sim D, z\sim q_\phi(z|c)}\left[\log p_\varphi(y|z)\right] \quad (8)$$

Reconstruction-based methods (Li et al., 2024) train the context encoder by reconstructing the dynamics and reward model with a decoder $p_\varphi(r, s'|s, a, z)$. The objective is to maximize the likelihood of the reconstructed dynamics and reward model:

$$\mathcal{L}_{\text{rec}} = \mathbb{E}_{(s,a,r,s')\sim D, z\sim q_\phi(z|c)}\log p_\varphi(r, s'|s, a, z) \quad (9)$$

Contrastive learning-based methods (Le-Khac et al., 2020) train the context encoder using contrastive losses, including InfoNCE (Oord et al., 2018), FOCAL (Li et al., 2020b) etc. These objectives minimize the distance between query and positive samples while maximizing the distance between query and negative samples. One commonly used contrastive loss (Li et al., 2020b) :

$$\mathcal{L}_{\text{cont}} = \mathbf{1}\{y_i = y_j\}\|z_i - z_j\|_2^2 \\ + \mathbf{1}\{y_i \neq y_j\}\frac{\beta}{\|z_i - z_j\|_2^n + \epsilon} \quad (10)$$

To integrate context uncertainty, the above learning objectives are modified by incorporating an uncertainty estimation network $h_\psi(\sigma|c)$. As shown in Figure 2 (a), given an input context $c$, the task representation $z$ is inferred using the context encoder $q_\phi(z|c)$, and a task representation learning objective $\mathcal{L}_\phi$ is computed. Simultaneously, a context uncertainty $\sigma_\psi$ is predicted by $h_\psi(\sigma|c)$. The heteroscedastic

loss $\mathcal{L}_{\phi,\psi}$ is then formulated by combining $\mathcal{L}_\phi$ with $\sigma_\psi$:

$$L_{(\phi,\psi)} = \frac{L_\phi}{\sigma_\psi^2} + \log \sigma_\psi \qquad (11)$$

The context uncertainty $\sigma_\psi$ is positively correlated with the task representation learning loss $\mathcal{L}_\phi$; the detailed proof is provided in Appendix A. This correlation implies that $\sigma_\psi$ can serve as a reliable measure of context uncertainty, as a higher task representation learning loss $\mathcal{L}_\phi$ typically indicates that the context is more likely to be ambiguous or OOD. Moreover, the heteroscedastic loss formulation provides an additional benefit: it adaptively adjusts the effective learning signal for $\mathcal{L}_\phi$, allowing the model to place less emphasis on uncertain contexts and more on confident ones, thereby improving the overall robustness of task representation learning.

## 4.2. Uncertainty-aware Policy Learning

### 4.2.1. META-POLICY LEARNING

The meta-policy $\pi_\theta(a|s,z)$ in COMRL is conditioned on the task representation $z$ to maximize the expected return across tasks. Its theoretical foundation lies in the framework of a partially observable Markov decision process (POMDP). Once the task representation $z$ is inferred from the context, the tuple $(s,z)$ is treated as a fully observable state, effectively transforming the POMDP into a Markov decision process (MDP). This enables the meta-policy in COMRL to be optimized using standard offline RL algorithms, such as CQL (Kumar et al., 2020), IQL (Kostrikov et al., 2021), BRAC (Wu et al., 2019), and TD3BC (Fujimoto & Gu, 2021).

The task representation $z$ in the meta-policy is typically derived as the mean of task representations of the context in existing COMRL methods. Nevertheless, assigning equal importance to all transitions within the context is vulnerable to ambiguous and OOD contexts. To address this limitation, we propose leveraging the uncertainty $\sigma$ estimated for each transition in the context $c$, and computing a weighted sum of each $z$ as the task representation. This approach reduces the influence of uncertain transitions, thereby improving the accuracy of task inference.

As illustrated in Figure 2 (b), given a context $c$ with $n$ transition tuples, the context encoder $q_\phi(z|c)$ is used to extract task representations for each transition, yielding $\{z_i|i=1,\ldots,n\}$. The uncertainties for each transition $\{\sigma_i|i=1,\ldots,n\}$ are then predicted by the uncertainty estimation network $h_\psi(\sigma|z)$. These uncertainties are then transformed into weights $\{w_i|i=1,\ldots,n\}$ via a softmax function. Finally, $z_w$ is computed as the weighted sum of $\{z_i|i=1,\ldots,n\}$.

$$z_w = \Sigma_{i=1}^n w_i z_i \qquad (12)$$

### 4.2.2. CONTEXT COLLECTING POLICY LEARNING

In the one-shot adaptation setting, collecting an effective context for task inference is crucial. Existing methods are not applicable as they either assume that adaptation contexts are either collected under the same behavior policy as the offline dataset or allow for multiple collection attempts using an exploratory policy.

To address these limitations, we propose training a context collecting policy that actively explore low-uncertainty contexts, thereby avoiding ambiguous and OOD contexts.

As illustrated in Figure 2 (b), the context collecting policy is defined as $\pi_\theta(a|s,z_0)$, where $z_0$ represents a zero-initialized task representation. The policy $\pi_\theta(a|s,z_0)$ is trained to maximize the action value function $Q_\theta(s,a,z_0)$, while incorporating an uncertainty penalty. This encourages the collection of informative and reliable contexts that improve task inference and enhance downstream adaptation performance:

$$J(\pi_\theta(a|s,z_0)) = \max \mathbb{E}_{(s,a,r,s')\sim D, \sigma\sim h_\psi(\sigma|q_\phi(c))}\Big[(r - \alpha\sigma)$$
$$+ \gamma\mathbb{E}_{a'\sim\pi_\theta(s,a,z_0)}Q_\theta(s',a',z_0)\Big] \qquad (13)$$

where $\alpha$ is a hyperparameter to balance the reward and the uncertainty penalty and $\gamma$ is the discount factor.

## 4.3. Uncertainty-aware One-shot Adaptation

As illustrated in Figure 2 (c), during the online adaptation, we employ the proposed context collecting policy $\pi_\theta(a|s,z_0)$ to collect one trajectory as adaptation context $c'$ in the unseen task $\mathcal{M}'$:

$$c' = \{(s_1,a_1,r_1,s_2), (s_2,a_2,r_2,s_3), \ldots,$$
$$(s_T,a_T,r_T,s_{T+1})\} \sim \langle\mathcal{M}', \pi_\theta(a|s,z_0)\rangle \qquad (14)$$

Since the ambiguous and OOD context cannot be fully prevented, we then utilize the context encoder $q_\phi$ and the uncertainty estimation network $h_\psi$ to infer the task representation $z'$, while restraining the influence of uncertain contexts in $c'$:

$$z' = \sum_{t=1}^T w_t z_t$$
$$\text{where} \quad z_t \sim q_\phi(z_t|s_t,a_t,r_t,s_{t+1}) \qquad (15)$$
$$w_t = \frac{\exp(-h_\psi(\sigma_t|z_t))}{\sum_{t=1}^T \exp(-h_\psi(\sigma_t|z_t))}$$

Finally, the adapted policy $\pi_\theta(a|s,z')$ for the unseen task $\mathcal{M}'$ is obtained by conditioning on the weighted task representation $z'$.

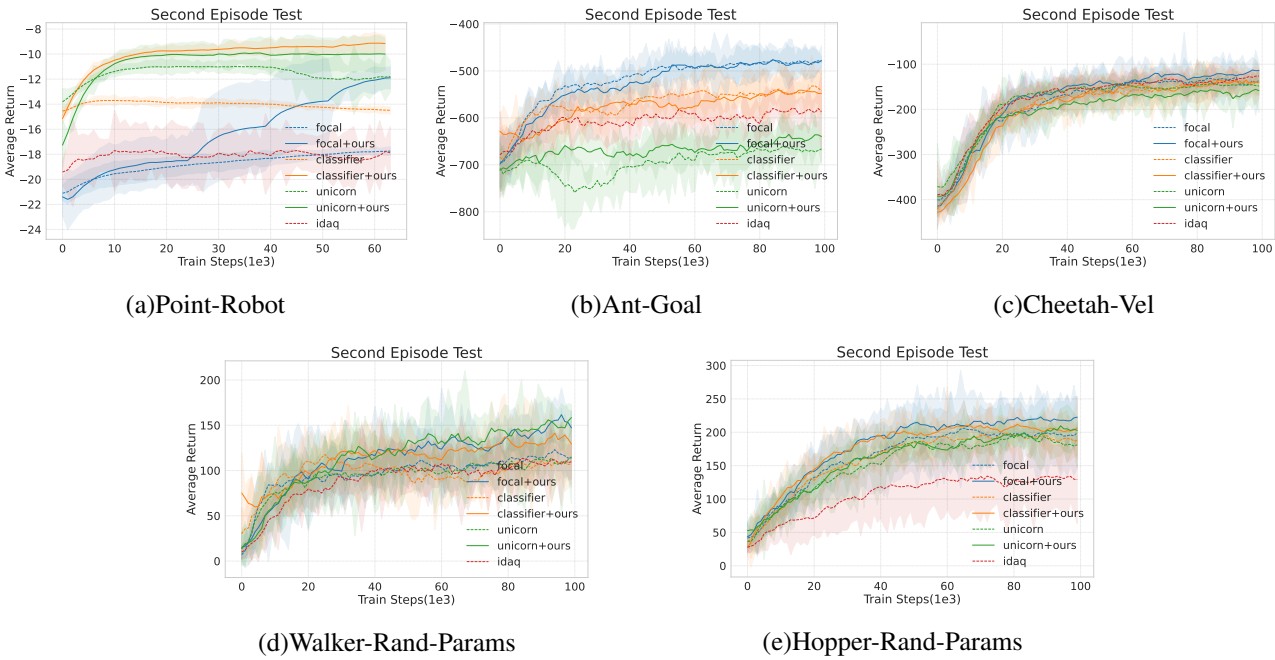

*Figure 3.* Average return of one-shot in five environments. Each curve represents the average return of four seeds.

## 5. Experiments

In our experiments, we aim to demonstrate the effectiveness of CERTAIN in online one-shot adaptation. Additionally, we conduct an ablation study to assess the necessity of both the uncertainty-aware context collecting policy and the uncertain context restraining mechanism.

### 5.1. Setup

**Environments.** We evaluate CERTAIN on one toy environment, Point-Robot, and four MuJoCo environments: Ant-Goal, Half-Cheetah-Vel, Walker-Rand-Params, and Hopper-Rand-Params.

The Point-Robot environment consists of an agent starting at $(0,0)$ and a goal positioned on a semicircle with a radius of 1. Tasks are distinguished by different goal positions. This environment enables intuitive visualization of agent trajectories and facilitates analysis of our method's underlying mechanism.

The MuJoCo environments present more complex and challenging scenarios. In Ant-Goal and Half-Cheetah-Vel, tasks are distinguished by different reward functions. In Walker-Rand-Params and Hopper-Rand-Params, tasks vary based on differences in environment dynamics parameters.

**Baselines.** We integrate CERTAIN with three baseline methods: CLASSIFIER (Zhang et al., 2025), FOCAL (Li et al., 2020b), and UNICORN (Li et al., 2024).

- CLASSIFIER (classifier-based) learns context representation by predicting task labels from the context using a cross-entropy loss.

- FOCAL (contrastive learning-based) learns context representations using a contrastive metric loss.

- UNICORN (reconstruction-based) extends FOCAL by incorporating a reconstruction loss.

Additionally, we compare with IDAQ (Wang et al., 2023a), which builds upon FOCAL and explicitly consider the OOD problem in online adaptation. The complete experimental configurations are summarized in Appendix B.

**Evaluation Metrics.** We evaluate CERTAIN in the one-shot online adaptation setting. Each evaluation consists of collecting two episode trajectories:

- First episode (context collection phase): The trajectory is collected using the context collecting policy, serving as the one-shot context.

- Second episode (adaptation phase): The trajectory is collected using the adapted policy, conditioned on the task representation $z$ inferred from the first episode's context.

For each evaluation, we run experiments across four random seeds and report the average return from the second episode as the final performance metric.

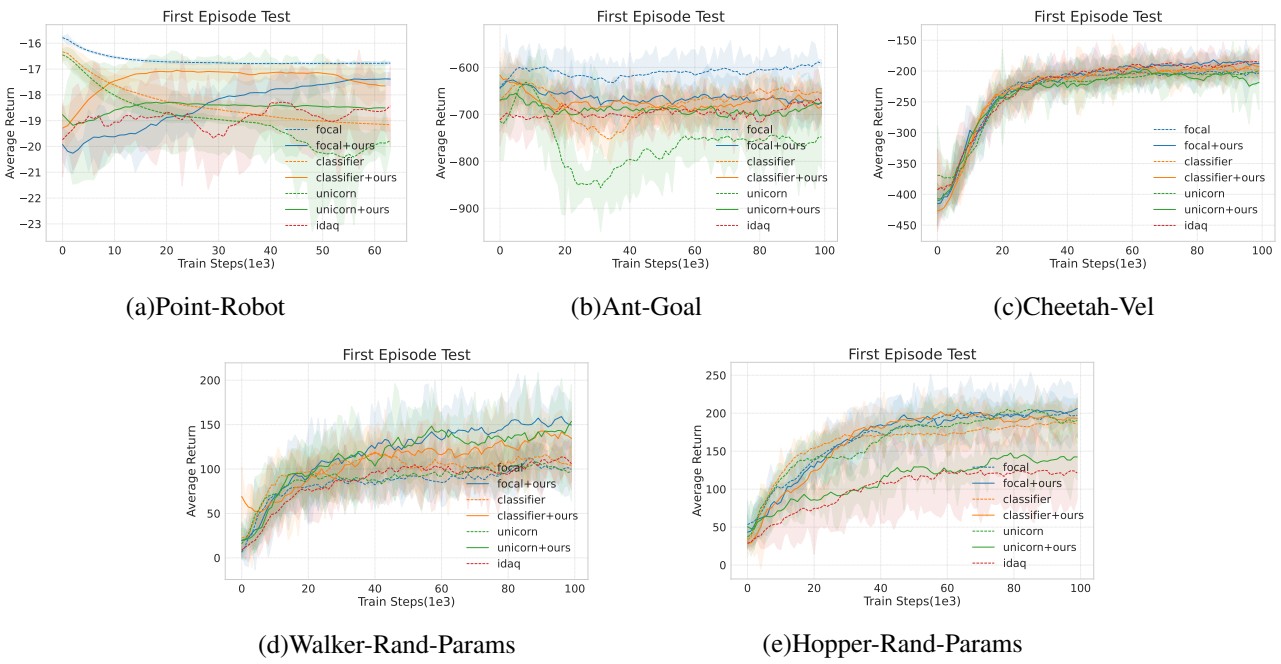

*Figure 4.* Average return of zero-shot in five environments.

## 5.2. Main Results

### 5.2.1. COMPARISON WITH BASELINES

We incorporate CERTAIN into three baseline methods and evaluate their one-shot adaptation performance against the original baselines. As shown in Figure 3, CERTAIN consistently matches or outperforms the baseline methods across all environments. Notably, CERTAIN+FOCAL consistently outperforms IDAQ (Wang et al., 2023a). The most significant improvement is observed in the Point-Robot environment, where CERTAIN+CLASSIFIER achieves a 36.9% performance improvement over the original CLASSIFIER.

Additionally, we evaluate zero-shot performance based on the return of the first episode trajectory, as shown in Figure 4. The results demonstrate that CERTAIN tends to outperform baseline methods in most environments. We attribute this to CERTAIN's context collecting policy, which gathers an initial trajectory with lower uncertainty, often resulting in higher returns.

Interestingly, FOCAL achieves higher zero-shot returns in the Point-Robot and Ant-Goal environments. We hypothesize that this is due to contrastive learning enabling FOCAL to learn a more balanced initial latent representation $z_0$, allowing the policy $\pi_\theta(s, z_0)$ to explore diverse goals more effectively. In contrast, CERTAIN's $z_0$ policy prioritizes collecting low-uncertainty transitions, which does not necessarily correlate with higher returns in the zero-shot setting.

Moreover, we observe that the zero-shot performance of baseline methods tends to degrade as training progresses. We hypothesize that this is because the policy $\pi_\theta(s, z_0)$ in the baselines becomes increasingly conservative over time. To support this claim, we provide an interpretable visualization of this performance decline in the Point-Robot environment in Appendix C.

Additional results on online adaptation in training tasks are presented in Appendix D, and results using a fixed offline context are provided in Appendix E. While our primary focus is on one-shot adaptation—the most challenging case in the few-shot setting—we also evaluate CERTAIN in a few-shot setting, where the agent can collect multiple context trajectories and update the task representation $z$ after each episode. As shown in Appendix F, CERTAIN consistently outperforms the baselines, demonstrating its effectiveness in few-shot adaptation as well.

### 5.2.2. VISUALIZATION

To better understand the rationale behind CERTAIN, we visualize the uncertainty of each context transition in the Point-Robot offline dataset. As shown in the left of Figure 5, transition uncertainty is notably higher near the starting point (shaded in red), where transitions are likely to occur across multiple tasks. This overlap leads to context ambiguity, making it challenging for the model to reliably distinguish between different tasks. In contrast, transitions farther from the starting point exhibit more pronounced re-

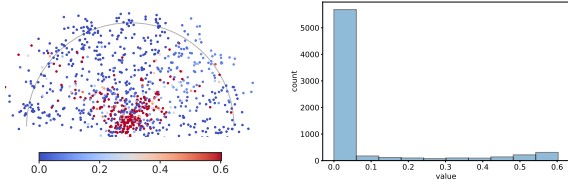

*Figure 5.* Context Uncertainty of the Point-Robot offline dataset. The left shows the uncertainty distribution across the state space, while the right presents the histogram of uncertainty values.

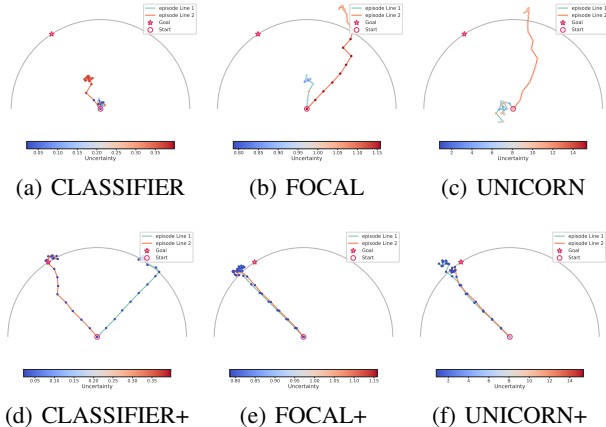

*Figure 7.* Visualization of two episode trajectories in the Point-Robot during evaluation. '+' denotes CERTAIN.

ward differences, facilitating task identification and resulting in lower uncertainty (shaded in blue). To characterize the uncertainty distribution, we present a histogram of all transitions on the right of Figure 5, showing that most transitions have low uncertainty, while highly uncertain transitions become progressively less frequent.

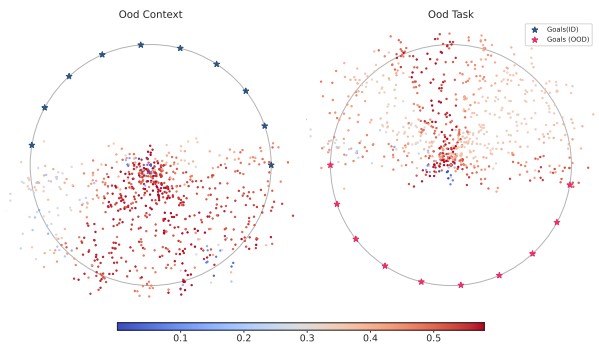

*Figure 6.* Context Uncertainty of OOD context and OOD task in the Point-Robot. The left shows the OOD context under in-distribution tasks, while the right presents the context uncertainty under OOD-task collected with behavior policy.

We evaluate CERTAIN's OOD-awareness by visualizing context uncertainty in the Point-Robot under two settings: OOD context and OOD task. In the OOD context setting, tasks remain the same as in training, but transitions are collected from the bottom semicircle, while the offline dataset comes from the top. In the OOD task setting, transitions are collected from the top semicircle using the same behavior policy in offline dataset, but with goals located in the bottom semicircle—unseen during training. As shown in Figure 6, both OOD settings exhibit significantly higher context uncertainty than the in-distribution case, highlighting CERTAIN's ability to detect distribution shifts—crucial for robust one-shot adaptation.

To further demonstrate the effectiveness of CERTAIN, we visualize the online trajectories over two episodes for different methods in the Point-Robot environment. As shown in Figure 7, the first episode trajectory (green) collected by CERTAIN moves directly toward the semicircle where

the goal is likely located, whereas baseline methods tend to explore around the starting point. In the second episode (orange), CERTAIN's trajectory is more direct and accurate compared to the baselines. Moreover, the uncertainties along CERTAIN's trajectories are lower than those of the baseline methods, suggesting that CERTAIN effectively reduces context uncertainty.

### 5.3. Ablation Study

To assess the necessity of the uncertainty-aware collecting policy and the uncertain context restraining mechanism in CERTAIN, we conduct ablation studies on CLASSIFIER-CERTAIN. For the ablation of the uncertainty-aware collecting policy, we compare CERTAIN with two variants: CERTAIN-Random and CERTAIN-Naive-$z_0$. CERTAIN-Random collects the adaptation context trajectory using a random policy, whereas CERTAIN-Naive-$z_0$ employs the same collecting policy as CERTAIN but without training during the meta-training phase. For the ablation of the uncertain context restraining mechanism, we introduce another variant, CERTAIN-Mean, which computes the mean of the representations inferred from the context trajectory. As shown in Table 1, CERTAIN consistently outperforms CERTAIN-Random, CERTAIN-Naive-$z_0$, and CERTAIN-Mean across all environments. These results highlight the necessity of both the uncertainty-aware collecting policy and the uncertain context restraining mechanism in CERTAIN. The ablation study for FOCAL-CERTAIN and UNICORN-CERTAIN is provided in Appendix G.1.

To evaluate the impact of the uncertainty penalty coefficient $\alpha$ in Equation (13), we conduct an ablation study. We observe no clear correlation between performance and the value of $\alpha$, and hypothesize that the penalty should be scaled to match the magnitudes of both the reward and context uncertainty. Detailed results are provided in Appendix G.2.

*Table 1.* Average return and standard deviation of the second episode trajectory collected with CLASSIFIER-CERTAIN and its variants.

| Env | CERTAIN | Naive-$z_0$ | Random | Mean |
|---|---|---|---|---|
| Point-Robot | **-9.27** $\pm$ 0.69 | -13.30 $\pm$ 1.00 | -10.04 $\pm$ 0.78 | -9.89 $\pm$ 0.35 |
| Ant-Goal | **-539.37** $\pm$ 40.83 | -576.40 $\pm$ 52.04 | -583.9 $\pm$ 72.63 | -555.67 $\pm$ 68.17 |
| Cheetah-Vel | **-110.02** $\pm$ 15.21 | -173.90 $\pm$ 7.34 | -126.23 $\pm$ 15.63 | -133.31 $\pm$ 33.38 |
| Walker | **144.40** $\pm$ 26.19 | 105.66 $\pm$ 13.26 | 100.99 $\pm$ 44.19 | 113.16 $\pm$ 37.07 |
| Hopper | **223.45** $\pm$ 28.23 | 202.22 $\pm$ 13.64 | 213.47 $\pm$ 20.01 | 206.6 $\pm$ 36.15 |

We also study the effect of offline dataset quality on CERTAIN's performance in the Point-Robot environment. Results show that performance improves most when the dataset is of medium quality. We hypothesize that extreme dataset qualities impair uncertainty estimation, resulting in suboptimal performance. Full results are presented in Appendix H.

## 6. Conclusion and Limitations

In this paper, we address the challenges of context ambiguity and OOD issues in one-shot adaptation for COMRL. By leveraging context uncertainty learning, we effectively identify ambiguous and OOD contexts. To further enhance task inference accuracy during one-shot online adaptation, we design an uncertainty-aware context collecting policy and an uncertain context restraining mechanism. Experimental results demonstrate that our method outperforms baseline approaches in both task inference accuracy and its ability to handle ambiguous and OOD contexts.

Despite its effectiveness, our method has certain limitations. First, it does not address the issue of spurious correlations, a common challenge in COMRL. However, it can be integrated with existing approaches, such as relabeling the context with task-specific models, to mitigate this problem. Second, while our method exhibits zero-shot adaptation capabilities, we attribute this primarily to the fact that our context collecting policy tends to gather low-uncertainty context trajectories, which often yield higher returns. Third, the context uncertainty estimated by our method can capture the presence of both OOD and ambiguous contexts, but it cannot explicitly distinguish between them. In future work, we plan to investigate techniques for disentangling OOD and ambiguous contexts and explore integrating our method with approaches designed to handle spurious correlations.

## Acknowledgments

The work is supported by the National Key Research and Development Program of China (No. 2020YFA0711402).

## Impact Statement

This paper presents work whose goal is to advance the field of Machine Learning. There are many potential societal consequences of our work, none which we feel must be specifically highlighted here.

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

# A. The Proof of Context Uncertainty

The modified heteroscedastic loss is defined as:

$$L_{(\phi,\psi)} = \frac{L_\phi}{\sigma_\psi^2} + \log \sigma_\psi \tag{16}$$

where $L_\phi$ denotes the task representation learning loss which is non-negative, and $\sigma_\psi$ represents the context uncertainty predicted by the network.

Since the neural network outputs the log-variance $o_\psi = \log \sigma_\psi$, we can reparameterize the loss as:

$$L_{(\phi,\psi)} = \frac{L_\phi}{\exp(2o_\psi)} + o_\psi \tag{17}$$

We then compute the first- and second-order derivatives of the loss with respect to $o_\psi$:

$$\frac{\partial L_{(\phi,\psi)}}{\partial o_\psi} = -\frac{2L_\phi}{\exp(2o_\psi)} + 1 \tag{18}$$

$$\frac{\partial^2 L_{(\phi,\psi)}}{\partial o_\psi^2} = \frac{4L_\phi}{\exp(2o_\psi)} \geq 0 \tag{19}$$

Since $L_\phi \geq 0$, the second derivative is non-negative, implying that the loss is convex with respect to $o_\psi$. Setting the first derivative to zero yields the minimizer $o_\psi^*$:

$$o_\psi^* = \frac{1}{2} \log(2L_\phi) \tag{20}$$

and consequently, the optimal context uncertainty is:

$$\sigma_\psi^* = \sqrt{2L_\phi} \tag{21}$$

This derivation demonstrates that the optimal context uncertainty $\sigma_\psi^*$ is positively correlated with the task representation learning loss $L_\phi$, thus justifying the use of $\sigma_\psi$ as a proxy for context uncertainty.

# B. Experimental Details

*Table 2.* Configurations and hyper-parameters used in the training process.

| Configurations | Point-Robot | Ant-Goal | Cheetah-Vel | Walker-Rand-Param | Hopper-Rand-Param |
|---|---|---|---|---|---|
| dataset size | $1e3$ | $1e6$ | | | |
| training steps | 60k | 200k | | | |
| test time context size | 1 trajectory (20 steps) | 1 trajectory (200 steps) | | | |
| unicorn weight $\frac{\eta}{1-\alpha}$ | 0.5 | 0.1 | 0.5 | 0.5 | 0.5 |
| punish weight(focal) | 10 | 0.1 | 2 | 5 | 0.1 |
| punish weight(classifier) | 10 | 5 | 10 | 25 | 10 |
| punish weight(unicorn) | 2 | 2 | 0.1 | 2 | 2 |
| context batch size | 1024 | 512 | 100 | 256 | |
| RL batch size | 256 | | | | |
| task representation dimension | 20 | | | | |
| learning rate | 3e-4 | | | | |
| RL network width | 256 | | | | |
| RL network depth | 3 | | | | |
| encoder width | 200 | | | | |
| encoder depth | 3 | | | | |
| uncertainty network width | 200 | | | | |
| uncertainty network depth | 1 | | | | |

## C. Explanation of Baseline Decline in Figure 4(a) with Training Steps

To further investigate generalization under the zero-shot condition, we examine how baseline performance evolves as training progresses. The following figures visualize the average return of FOCAL, CLASSIFIER, and UNICORN at various training steps ($\times 10^3$). In each subfigure, the trajectory labeled "Episode Line 1" corresponds to zero-shot evaluation. We observe that the average return of baselines tends to decline as training continues, suggesting a possible overfitting or memorization of training context.

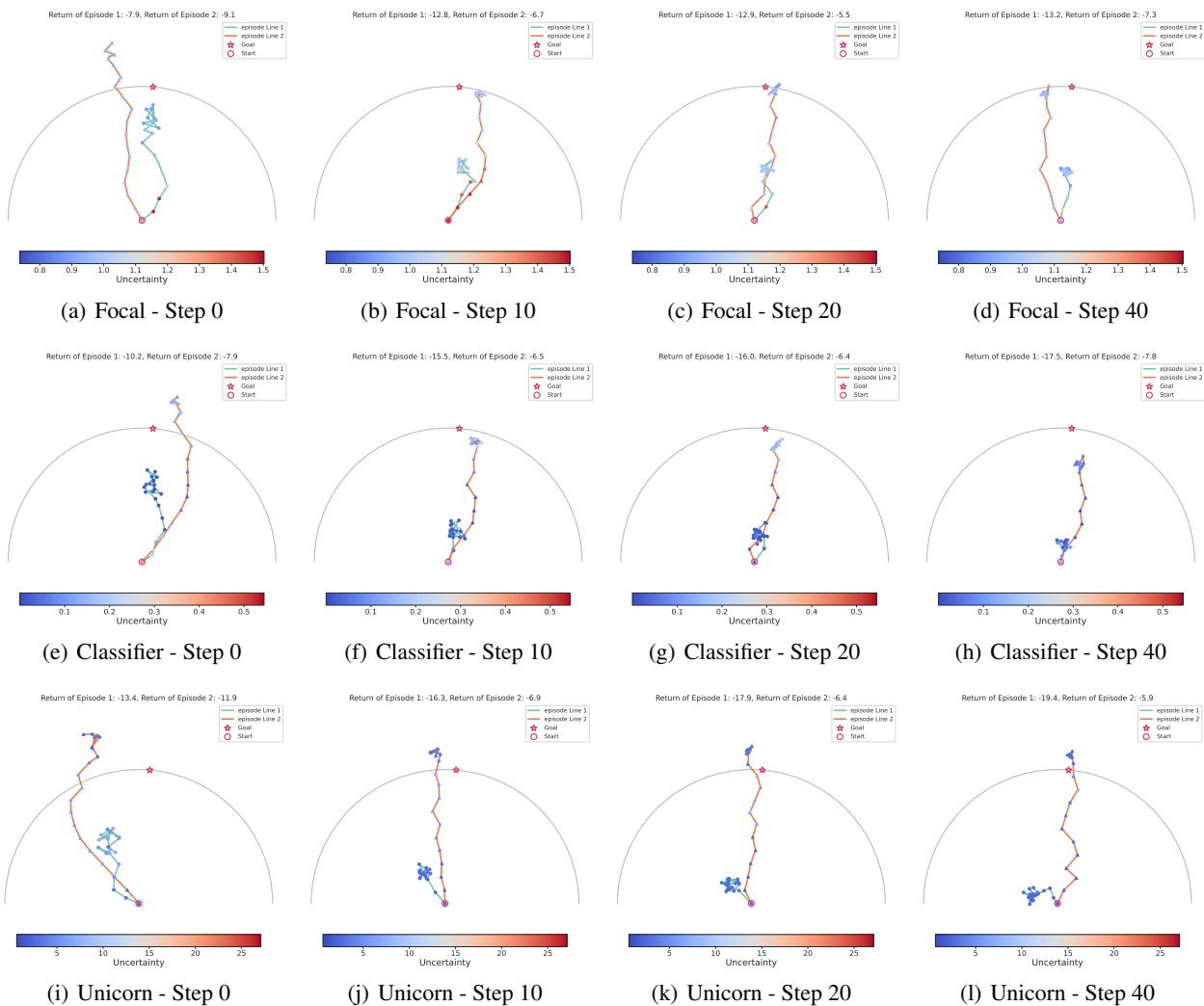

*Figure 8.* Average return of baseline methods under zero-shot conditions at different training steps ($\times 10^3$). Each trajectory labeled "Episode Line 1" corresponds to the zero-shot rollout.

# D. Online Adaptation Results on Training Tasks

## D.1. One-shot Results

We present the online adaptation results on the training tasks in Figure 9. The results indicate that CERTAIN consistently outperforms the baselines across almost all the environments, mirroring its performance on the testing tasks (Figure 3).

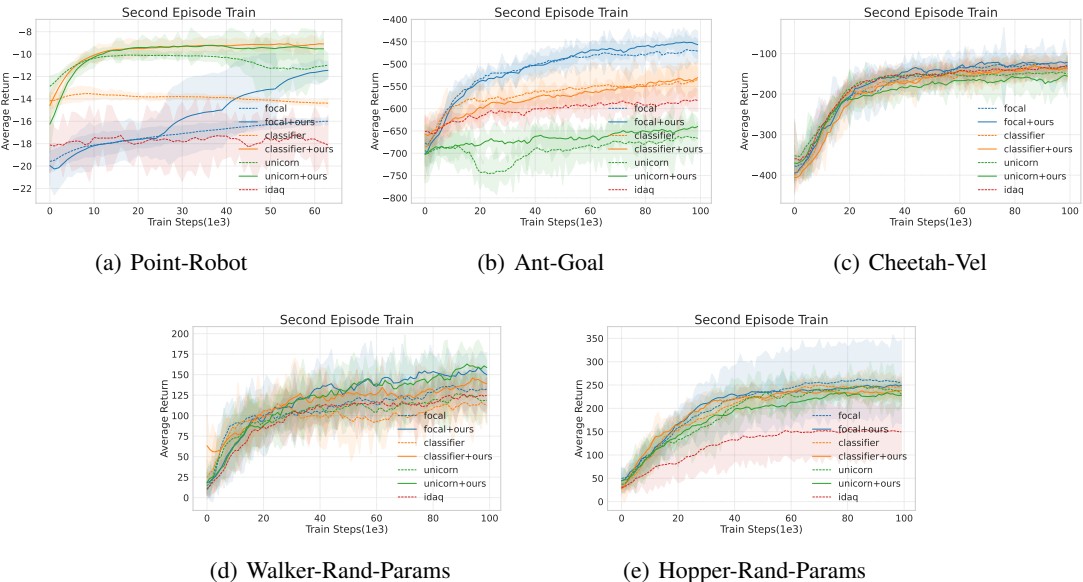

Figure 9. Average return of one-shot in five environments on training tasks. Each curve represents the average return of four seeds.

## D.2. Zero-shot Results

We also present the zero-shot results on the training tasks in Figure 10, which are consistent with those observed on the testing tasks in Figure 4.

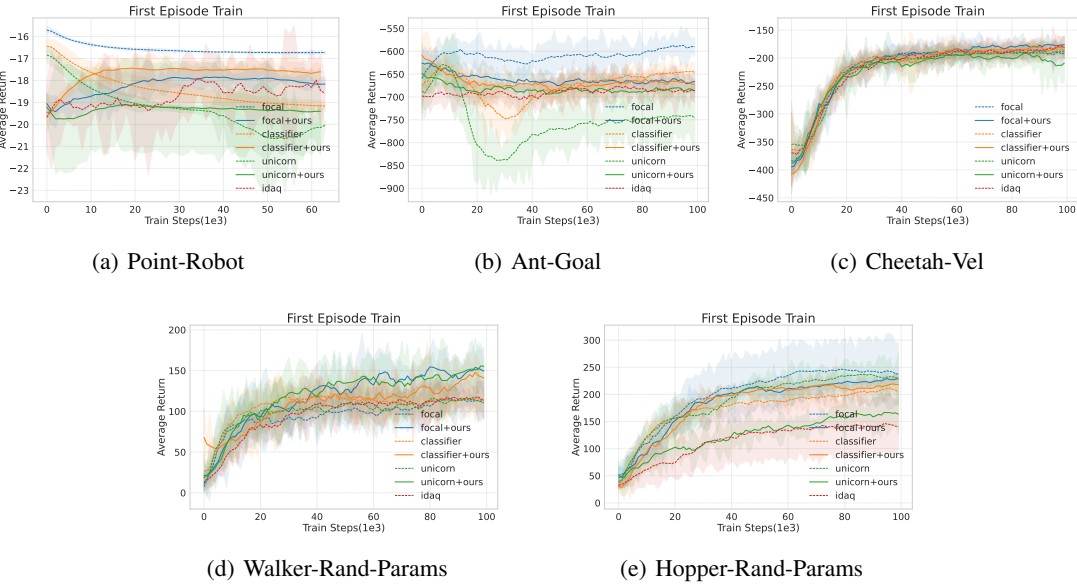

Figure 10. Average return of zero-shot in five environments on training tasks.

# E. Adaptation Results with Given Offline Context

Similar to other COMRL methods, we present adaptation results with a given offline context in Figure 11 and Figure 12.

## E.1. Results on Training Tasks

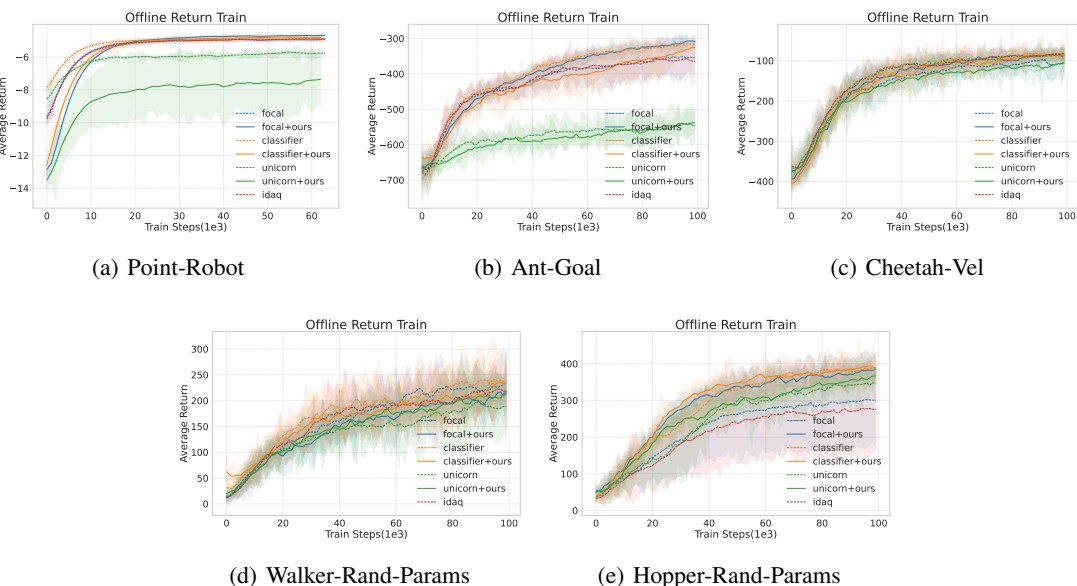

(a) Point-Robot        (b) Ant-Goal        (c) Cheetah-Vel

(d) Walker-Rand-Params        (e) Hopper-Rand-Params

*Figure 11.* Average return on the training tasks with given offline context in five environments. Each curve represents the average return of four seeds.

## E.2. Results on Testing Tasks

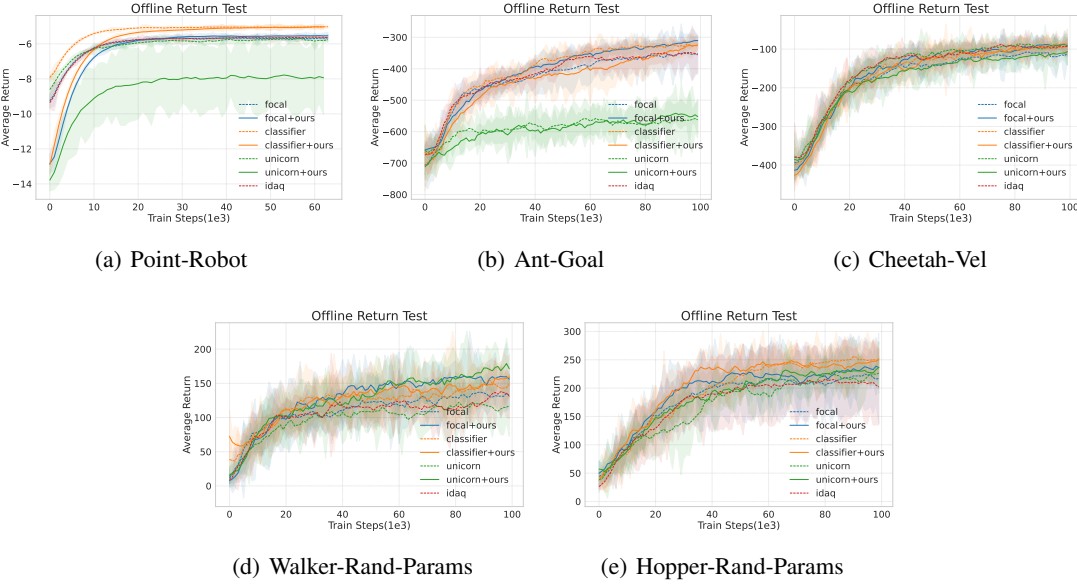

(a) Point-Robot        (b) Ant-Goal        (c) Cheetah-Vel

(d) Walker-Rand-Params        (e) Hopper-Rand-Params

*Figure 12.* Average return on testing tasks with given offline context in five environments.

## F. Few-shot Performance

In the main paper, a one-shot setting is employed; here we supplement the results under a few-shot setting. Under the few-shot setting, we randomly sample 5 episodes during evaluation. The table below presents the average return ($\pm$ standard deviation) for each episode, with the best return in each episode highlighted in bold.

*Table 3.* Few-shot average return and standard deviation across five sampled episodes in the Point-Robot environment. Bold values indicate the best performance per episode across all methods.

| Episode | FOCAL | FOCAL+Ours | CLASSIFIER | CLASSIFIER+Ours | UNICORN | UNICORN+Ours |
|---------|-------|------------|------------|-----------------|---------|--------------|
| Episode 1 | **-16.74**±0.08 | -17.40±0.28 | -19.13±0.26 | -17.66±0.71 | -19.49±0.47 | -18.52±1.73 |
| Episode 2 | -17.82±0.33 | -11.97±1.11 | -14.51±0.27 | **-8.92**±0.81 | -11.46±1.07 | -11.00±2.17 |
| Episode 3 | -18.13±0.24 | -12.34±1.24 | -12.79±1.01 | **-9.33**±0.59 | -13.26±1.95 | -9.53±0.79 |
| Episode 4 | -18.31±0.26 | -12.36±1.05 | -12.69±1.00 | **-9.64**±0.66 | -14.66±2.19 | -9.94±1.12 |
| Episode 5 | -18.48±0.23 | -12.35±0.98 | -13.14±1.01 | **-9.69**±1.00 | -15.07±1.68 | -10.53±0.61 |

## G. Ablation Study

### G.1. Ablation Results on FOCAL-CERTAIN and UNICORN-CERTAIN

In this section, we provide additional ablation results for FOCAL-CERTAIN and UNICORN-CERTAIN. We compare the performance of FOCAL-CERTAIN and UNICORN-CERTAIN with two variants: Naive-$z_0$ and Random. For detailed explanations, please refer to Section 5.3.

*Table 4.* Average return and standard deviation of the second episode trajectory collected with FOCAL-CERTAIN and its variants in the point-robot environment.

| Env | CERTAIN | Naive-$z_0$ | Random | Mean |
|-----|---------|-------------|--------|------|
| Point-Robot | **-11.20** $\pm$ 0.77 | -16.38 $\pm$ 0.54 | -12.52 $\pm$ 1.28 | -13.80 $\pm$ 2.63 |
| Ant-Goal | **-432.83** $\pm$ 12.98 | -531.40 $\pm$ 31.06 | -464.78 $\pm$ 23.65 | -459.03 $\pm$ 6.08 |
| Cheetah-Vel | -117.14 $\pm$ 31.13 | -137.00 $\pm$ 14.40 | **-90.37** $\pm$ 16.54 | -146.70 $\pm$ 43.84 |
| Walker | **139.98** $\pm$ 16.44 | 104.71 $\pm$ 18.89 | 112.27 $\pm$ 22.70 | 136.65 $\pm$ 34.17 |
| Hopper | **231.2** $\pm$ 22.51 | **234.45** $\pm$ 29.55 | 186.20 $\pm$ 14.79 | 192.70 $\pm$ 9.90 |

*Table 5.* Average return and standard deviation of the second episode trajectory collected with UNICORN-CERTAIN and its variants in the point-robot environment.

| Env | CERTAIN | Naive-$z_0$ | Random | Mean |
|-----|---------|-------------|--------|------|
| Point-Robot | **-10.06** $\pm$ 1.27 | -15.98 $\pm$ 2.20 | -15.81 $\pm$ 1.20 | -15.98 $\pm$ 2.20 |
| Ant-Goal | -637.00 $\pm$ 66.39 | **-618.93** $\pm$ 59.30 | -630.73 $\pm$ 25.91 | -660.45 $\pm$ 29.03 |
| Cheetah-Vel | -139.10 $\pm$ 39.70 | -147.38 $\pm$ 16.89 | **-117.02** $\pm$ 18.78 | -169.53 $\pm$ 22.23 |
| Walker | **186.35** $\pm$ 37.75 | 129.91 $\pm$ 60.48 | 148.63 $\pm$ 25.42 | 109.63 $\pm$ 37.53 |
| Hopper | 188.93 $\pm$ 10.11 | 197.83 $\pm$ 21.95 | 148.57 $\pm$ 4.38 | **225.47** $\pm$ 13.51 |

### G.2. Impact of Uncertainty Penalty $\alpha$ on Performance

In this section, we provide ablation results for the hyperparameter of uncertainty penalty $\alpha$, comparing the results for $\alpha \in \{0.1, 1, 10\}$.

*Table 6.* Average return and standard deviation of the **second episode** trajectory under different methods and uncertainty penalty settings

| Env | Classifier | | | Focal | | | Unicorn | | |
|---|---|---|---|---|---|---|---|---|---|
| $\alpha$ | 0.1 | 1 | 10 | 0.1 | 1 | 10 | 0.1 | 1 | 10 |
| Point-Robot | -9.53±0.94 | -9.58±0.70 | **-9.27**±0.69 | -12.54±1.70 | -15.47±5.48 | **-11.20**±0.77 | -13.09±4.83 | -13.07±5.69 | **-9.68**±1.35 |
| Ant-Goal | -527.4±34.86 | **-519.68**±48.66 | -530.6±40.89 | **-432.83**±12.98 | -451.23±18.39 | -500.28±43.87 | -649.13±61.54 | -665.48±84.17 | **-633.4**±30.92 |
| Cheetah-Vel | -129.18±33.15 | -164.53±49.16 | **-110.02**±15.21 | -114.98±40.86 | -141.74±38.00 | **-106.9**±18.94 | **-139.10**±39.70 | -175.95±45.61 | -195.55±40.24 |
| Walker | **128.21**±30.67 | 106.02±41.56 | 83.23±31.72 | 100.32±25.87 | **186.23**±39.37 | 132.95±16.20 | **168.32**±54.78 | 117.51±27.72 | 97.77±34.94 |
| Hopper | **232.48**±25.71 | 193.9±16.31 | 223.45±28.23 | **231.2**±22.51 | 194.30±101.60 | 194.77±12.97 | 174.17±56.26 | **183.99**±80.66 | 140.03±15.83 |

*Table 7.* Average return and standard deviation of the **first episode** trajectory under different methods and uncertainty penalty settings

| Env | Classifier | | | Focal | | | Unicorn | | |
|---|---|---|---|---|---|---|---|---|---|
| $\alpha$ | 0.1 | 1 | 10 | 0.1 | 1 | 10 | 0.1 | 1 | 10 |
| Point-Robot | **-17.6**±0.91 | -17.63±0.17 | -17.68±0.49 | **-17.32**±0.13 | -17.70±0.80 | -11.57±0.76 | **-17.58**±0.56 | -17.95±1.32 | -20.44±3.57 |
| Ant-Goal | -668.38±53.85 | **-639.47**±20.72 | -662.13±20.72 | -651.70±38.44 | -671.25±36.89 | **-615.525**±16.73 | -694.7±3.49 | **-672.18**±24.88 | -698.4±61.56 |
| Cheetah-Vel | -183.68±40.76 | -173.73±19.21 | **-172.05**±17.91 | -193.35±17.59 | **-180.78**±22.01 | -186.88±17.26 | -210.13±17.55 | **-181.83**±22.10 | -219.18±51.25 |
| Walker | 131.65±36.93 | **138.75**±28.10 | 122.04±32.37 | 101.57±39.44 | **185.08**±48.90 | 137.25±25.46 | **179.73**±61.36 | 148.0125±49.78 | 37.74±33.10 |
| Hopper | 203.28±13.20 | **219.85**±11.28 | 199.95±9.89 | **194.48**±14.13 | 165.22±80.05 | 180.03±8.25 | 140.7±54.38 | **166.3**±24.82 | 34.05±33.90 |

## H. Performance Results under Low, Mid, and Expert Offline Data

While the main paper adopts the *mix* offline dataset, this section complements the results by evaluating our method under **low**, **mid**, and **expert** quality offline datasets in the Point-Robot environment.

We visualize the zero-shot (left) and one-shot (right) performance across different offline data quality levels. The solid lines represent our method, and the dashed lines denote the corresponding baselines. All results are evaluated every 1000 training steps.

**Data Quality Definitions:**

- **Low-quality data** contains high levels of noise and inaccuracies, often due to environment randomness or poor dynamics modeling, leading to unstable training and higher uncertainty.

- **Mid-quality data** contains moderate noise and some useful information for generalization, providing a balanced trade-off between reliability and diversity.

- **Expert-level data** is high-quality data with minimal noise, curated by expert policies. It is reliable but may contain spurious correlations due to lack of exploration diversity.

### H.1. Low Offline Dataset

Our method improves over FOCAL and UNICORN baselines, indicating robustness to noisy data. However, it slightly underperforms the CLASSIFIER baseline, suggesting that noise and incompleteness can hinder effectiveness. This highlights the need for additional noise handling strategies in future work.

### H.2. Mid Offline Dataset

Across all baselines, our method consistently improves second episode performance, demonstrating its effectiveness in moderately reliable settings and achieving state-of-the-art results under mid-quality data.

### H.3. Expert Offline Dataset

Despite high data quality, both baselines and our method perform worse than on mid or mix datasets. Visualizations suggest that expert data may introduce spurious correlations due to limited exploration, which hinders robust task inference.

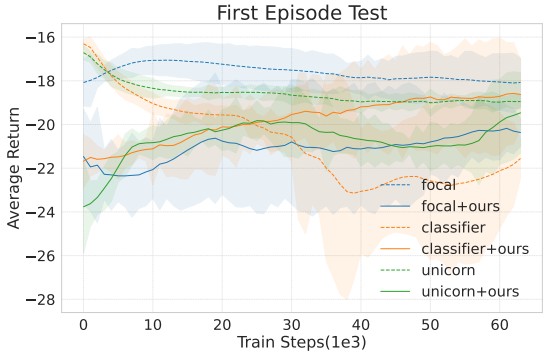

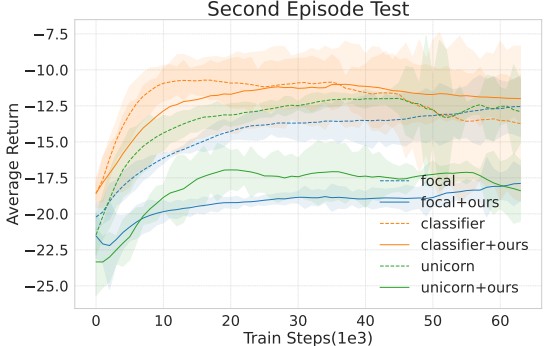

(a) Zero-shot on Low-quality Dataset

(b) One-shot on Low-quality Dataset

*Figure 13.* Performance under low-quality offline data.

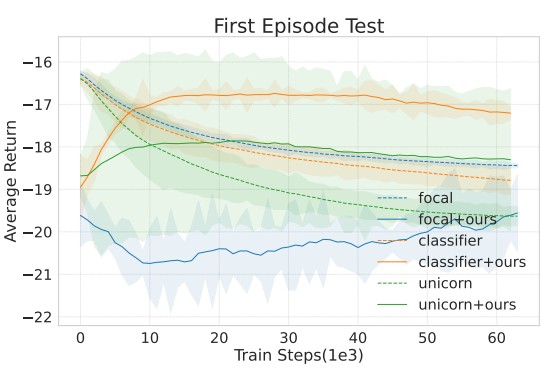

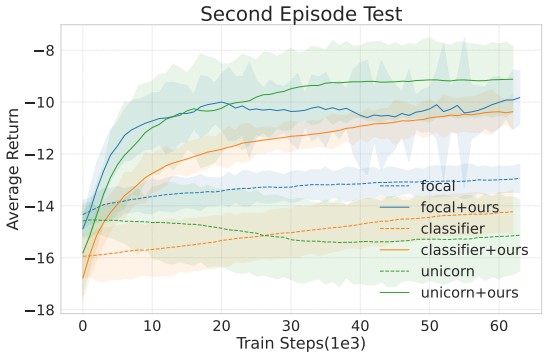

(a) Zero-shot on Mid-quality Dataset

(b) One-shot on Mid-quality Dataset

*Figure 14.* Performance under mid-quality offline data.

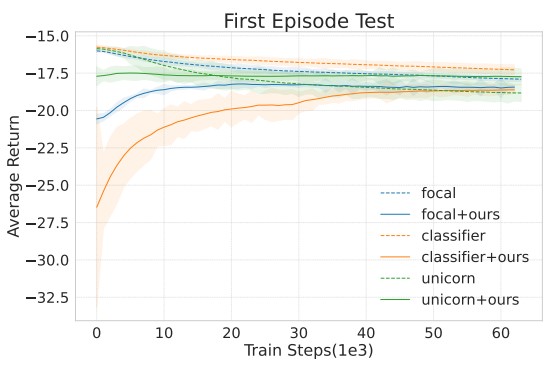

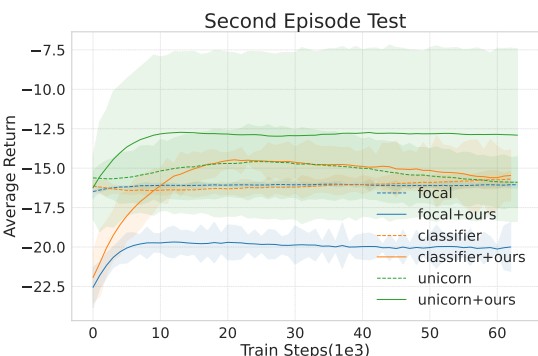

(a) Zero-shot on Expert-quality Dataset

(b) One-shot on Expert-quality Dataset

*Figure 15.* Performance under expert-level offline data.

