# OpenReview forum: "CERTAIN: Context Uncertainty-aware One-Shot Adaptation for Context-based Offline Meta Reinforcement Learning"
_ICML.cc/2025/Conference — ICML 2025 poster_

### Official Review · Reviewer_eS1K · 2025-02-26

**Overall Recommendation:** 3

**Summary:**

The paper presents CERTAIN, a novel framework designed to address challenges in context-based offline meta-reinforcement learning (COMRL), particularly context ambiguity and out-of-distribution (OOD) issues, in one-shot adaptation settings. The authors propose leveraging heteroscedastic uncertainty in task representation learning to identify and mitigate the negative impact of ambiguous and OOD contexts on task inference during online adaptation.
## update after rebuttal
Thanks for the author's reply, I have no further questions.

**Claims And Evidence:**

Yes

**Essential References Not Discussed:**

No

**Experimental Designs Or Analyses:**

The experimental design is basically reasonable.

**Methods And Evaluation Criteria:**

Yes

**Other Comments Or Suggestions:**

No other suggestions.

**Other Strengths And Weaknesses:**

**Strengths**

1.The proposed integration of heteroscedastic uncertainty into task representation learning for COMRL is a novel approach that addresses both context ambiguity and OOD issues, which have been insufficiently explored in prior research.

2. The paper effectively identifies a practical problem in one-shot adaptation settings and introduces a solution with direct real-world relevance, improving sample efficiency and safety in reinforcement learning tasks.

3. Extensive empirical evaluations across multiple environments, including toy tasks and complex MuJoCo environments, demonstrate the robustness and effectiveness of the CERTAIN framework.

4. CERTAIN is designed as a plug-in framework that can be easily integrated into existing COMRL methods (e.g., classifier-based, reconstruction-based), making it adaptable to various contexts and methods.

**Weaknesses**

1.While the paper provides strong empirical results, it lacks formal theoretical analysis or guarantees regarding the performance of the uncertainty-aware components, which would strengthen the theoretical foundation.

2.While the paper presents ablation studies on certain components, further detailed analysis on the interplay between the different uncertainty mechanisms (e.g., uncertainty estimation network and context collection policy) could provide a deeper understanding of their individual contributions.

**Questions For Authors:**

From my perspective, methods like Algorithm Distillation handle both task inference and context-conditioned policy within a single model, such as a Transformer. Compared to these methods, what are the advantages and disadvantages of the authors' approach, which uses multiple models for learning?

**Relation To Broader Scientific Literature:**

The key contributions of this paper build on the context-based offline meta-reinforcement learning (COMRL) framework, addressing the limitations of context ambiguity and OOD issues, which have been largely overlooked in prior COMRL methods. The integration of heteroscedastic uncertainty in task representation learning and context collection policies extends existing approaches, such as classifier-based and contrastive learning methods, by improving adaptation robustness in one-shot settings.

**Theoretical Claims:**

No proofs for theoretical claims.

---

> ### Author Rebuttal · Authors · 2025-03-31
>
> We sincerely appreciate the reviewers’ constructive feedback. Below, we respond to each concern point by point. We will incorporate all the reviewers’ suggestions in the final version of the paper.
>
> > Reviewer:
> >
> > Weaknesses
> >
> > 1.While the paper provides strong empirical results ...
> >
> > 2.While the paper presents ablation studies on certain components ...
>
> Response:
> 1. We acknowledge the lack of a theoretical analysis and plan to explore this aspect further in future work. However, our primary contribution lies in addressing uncertainty in few-shot (specifically one-shot) adaptation, which we hope will open new avenues for research in the OMRL field.
> 2. To further validate our approach, we have added multiple experiments in the anonymous link, demonstrating the ability of the uncertainty estimation network to distinguish OOD contexts at <https://anonymous.4open.science/r/CERTAIN-6073/experiment1.md> and highlighting the importance of the context collection policy for few-shot performance at <https://anonymous.4open.science/r/CERTAIN-6073/experiment3.md>.
>
>
> > Reviewer: From my perspective, methods like Algorithm Distillation handle both task inference and context-conditioned policy within a single model, such as a Transformer. Compared to these methods, what are the advantages and disadvantages of the authors' approach, which uses multiple models for learning?
>
> Response:
> Using a single model for both task inference and policy execution typically requires a massive dataset and a large number of parameters. Moreover, these two fundamentally different objectives can interfere with each other’s learning, making optimization more challenging. In contrast, our approach separates these processes into multiple models, reducing learning difficulty while enhancing interpretability and controllability.

---

> > ### Comment · Reviewer_eS1K · 2025-04-07
> >
> > Thanks for the author's reply, I have no further questions.

---

> > > ### Author Response · Authors · 2025-04-09
> > >
> > > Given that we addressed your primary concerns raised in the review, we would kindly ask you to adjust your review score while taking the rebuttal into account.

---

### Official Review · Reviewer_eruf · 2025-03-03

**Overall Recommendation:** 3

**Summary:**

This paper presents the CERTAIN method for Offline Meta Reinforcement Learning. The CERTAIN method models uncertainty for each transition sample explicitly, and performs task representation by selecting samples with lower uncertainty, thereby achieving more accurate task identification. Experimental results in MuJoCo environments show that CERTAIN, when combined with three different Offline Meta RL baselines, demonstrates superior reward performance.

**Claims And Evidence:**

See Strengths and Weaknesses.

**Essential References Not Discussed:**

See Strength and Weaknesses.

**Experimental Designs Or Analyses:**

See Strength and Weaknesses.

**Methods And Evaluation Criteria:**

See Strength and Weaknesses.

**Other Comments Or Suggestions:**

None

**Other Strengths And Weaknesses:**

This paper demonstrates certain advantages in addressing the problem of Offline Meta Reinforcement Learning:



- The paper clearly outlines two reasons for task identification failures in COMRL: Context Ambiguity and OOD Context.

- The proposed method is simple and straightforward.

- Experiments in the MuJoCo environment show that the CERTAIN method provides an improvement in reward performance.



However, there are several issues with the paper:



1. **Uncertainty Estimation**: The paper uses the Heteroscedastic Uncertainty loss function from regression tasks. While this loss function is theoretically well-supported for maximizing likelihood in regression tasks, it may not necessarily have the same properties in the context of representation learning as presented in this paper. Therefore, I am skeptical about the choice of this uncertainty estimation method. For example, when combining CERTAIN with classification methods, would it be more reasonable to use a classification-specific Heteroscedastic Uncertainty loss function? Additionally, what is the significance of such a loss function in contrastive learning?

2. **Context Collecting Policy Learning**: In the learning of the context collecting policy, the policy still needs to be learned through the reward r. But if there is no prior task, what is the reward r in this case? Furthermore, the uncertainty estimation also requires input from (s,a,s′,r), meaning that uncertainty should be related to the reward, which corresponds to the task itself. However, the context collecting policy should be a task-independent policy. How then can uncertainty be applied to the context collecting policy? For example, suppose there are three tasks with significantly different distributions. A given transition s,a,s′ may have low uncertainty when the reward r1 from task 1 is given, but be an out-of-distribution (OOD) sample when the rewards r2 or r3 from tasks 2 or 3 are provided. In this case, a sample that is valid for task 1 may become invalid for tasks 2 or 3. Hence, I don't find the uncertainty-based context collecting policy to be fully reasonable.

3. **Data Augmentation Methods**: The paper lacks a discussion or comparison with data augmentation methods in COMRL, such as MBML [1], COSTA [2], or ReDA [3]. Data augmentation techniques are likely to be effective in reducing uncertainty.

4. **Comparison with SOTA Methods**: While the paper mentions Mutual Information-based methods like CORRO [4] and CSRO [5] in contrastive learning, it does not compare CERTAIN with these methods. Given that these methods are likely more state-of-the-art than FOCAL, such a comparison would strengthen the paper.

5. **Narrow Focus on One-Shot In-Distribution Task Adaptation**: The paper primarily focuses on one-shot in-distribution task adaptation, which is a rather narrow setting. One-shot adaptation requires only a single trajectory to adapt to the task, but there are many few-shot methods, such as Thompson Sampling. Expanding the one-shot scenario to a few-shot setting and comparing it with few-shot adaptation methods could make the paper more solid. Additionally, many previous COMRL algorithms, including the baseline FOCAL and methods like CORRO, deal with OOD task adaptation, but this paper only experiments with in-distribution tasks.



If the issues above are effectively addressed, I would consider re-reviewing the paper.



[1] Multi-task Batch Reinforcement Learning with Metric Learning



[2] Cost-aware Offline Safe Meta Reinforcement Learning with Robust In-Distribution Online Task Adaptation



[3] Disentangling Policy from Offline Task Representation Learning via Adversarial Data Augmentation



[4] Robust task representations for offline meta-reinforcement learning via contrastive learning



[5] Context Shift Reduction for Offline Meta-Reinforcement Learning

**Questions For Authors:**

See Strength and Weaknesses.

**Relation To Broader Scientific Literature:**

See Strength and Weaknesses.

**Theoretical Claims:**

No theoretical claims in this paper.

---

> ### Author Rebuttal · Authors · 2025-03-31
>
> We sincerely appreciate the reviewers’ constructive feedback. Below, we respond to each concern point by point. We will incorporate all the reviewers’ suggestions in the final version of the paper.
>
> > Reviewer: Uncertainty Estimation: The paper uses the Heteroscedastic Uncertainty loss function from regression tasks ...
>
> Response:
> In this paper, the heteroscedastic loss is formulated as $\frac{L}{\sigma^2} + \log \sigma$, where the heteroscedastic term $\sigma$ captures the magnitude of the original loss L. Thus, as long as L is a positive-valued function—such as MSE, cross-entropy, or contrastive loss—the learning process of heteroscedasticity $\sigma$ is guaranteed. Moreover, heteroscedasticity $\sigma$ has already been applied to contrastive losses, as demonstrated in [1], where it was incorporated into the triplet contrastive loss.
> Regarding the role of heteroscedasticity $\sigma$ in contrastive loss, it can be interpreted as the separability of samples from different tasks given a specific encoder model.
>
> [1] Unsupervised Data Uncertainty Learning in Visual Retrieval Systems
>
> > Reviewer: Context Collecting Policy Learning: In the learning of the context collecting policy, the policy still needs to be learned through the reward r. But ...
>
> Response:
> 1. We consider only cases where the offline dataset includes rewards, meaning all transitions in the dataset are complete in the form of  $(s, a, r, s')$. For scenarios where the dataset lacks rewards, we acknowledge this as an important and interesting problem and plan to explore it in future work.
> 2. The training process of the context collection policy $\pi_\theta(s, z_0)$ is task-agnostic. Consequently, $\pi_\theta(s, z_0)$ focuses on overall uncertainty across the dataset distribution and cannot ensure that collected samples for a specific task have low uncertainty. To mitigate the impact of high-uncertainty samples, we employ an uncertainty-restraining method, which helps reduce task inference uncertainty and ultimately improves meta-policy performance.
>
> > Reviewer: Data Augmentation Methods: The paper lacks a discussion or comparison with data augmentation methods in COMRL, such as MBML [1], COSTA [2], or ReDA [3]. Data augmentation techniques are likely to be effective in reducing uncertainty.
>
> Response: Data augmentation can improve the robustness of the context encoder in OOD scenarios. However, our method focuses on identifying and evaluating ambiguous and potentially OOD contexts for adaptation purposes. Furthermore, since our approach is plug-in compatible, it can be integrated with methods that utilize data augmentation.
>
> > Reviewer: Comparison with SOTA Methods: While the paper mentions Mutual Information-based methods like CORRO [4] and CSRO [5] in contrastive learning, it does not compare CERTAIN with these methods ...
>
> Response: In our adopted baseline [2], a complete theoretical derivation of CORRO and CSRO, along with their performance evaluations, is provided within a unified framework, UNICORN, from a mutual information perspective. Therefore, we believe that conducting experiments based on UNICORN effectively represents CORRO and CSRO.
>
> [2] Towards an Information Theoretic Framework of Context-Based Offline Meta-Reinforcement Learning
>
> > Reviewer: Narrow Focus on One-Shot In-Distribution Task Adaptation: The paper primarily focuses on one-shot in-distribution task adaptation ...
>
> Response: We consider one-shot learning the most challenging case within few-shot settings. Achieving high performance with as few shots as possible better demonstrates the effectiveness of the adaptation algorithm. We acknowledge the reviewer’s suggestion regarding broader few-shot settings. Therefore, we have supplemented few-shot (5 episodes) experiments on the Point Robot task at <https://anonymous.4open.science/r/CERTAIN-6073/experiment3.md> and added experiments on OOD tasks (where goal is at lower semicircle) at <https://anonymous.4open.science/r/CERTAIN-6073/experiment4.md>.

---

### Official Review · Reviewer_a856 · 2025-03-09

**Overall Recommendation:** 3

**Summary:**

This paper studies the task representation problem in context-based offline meta-reinforcement learning (COMRL). It first identifies the problem of task uncertainty in a context, including task ambiguity and out-of-distribution problems. Then, the paper proposes a training method that learns both context representation and uncertainty and trains a policy that maximizes return while minimizing the estimated uncertainty. In experiments on MuJoCo tasks, the method outperforms baselines in both one-shot adaptation settings and zero-shot settings.

**Claims And Evidence:**

Most claims made in this paper are clear. But there are two issues:
- (line 40, right) "Prior methods often assume that contexts are either in-distribution or can be collected through multiple rounds." However, CORRO [1]  focuses on addressing OOD contexts in OMRL.
- Definitions 3.1, 3.2, 3.3 are more like an intuitive explanation and the definitions are either informal or not used in the following writing. Though, this can be a minor issue because the paper does not claim some theoretical contributions.

[1] Robust task representations for offline meta-reinforcement learning via contrastive learning. ICML 2022.

**Essential References Not Discussed:**

I did not find a missing citation. But I think some related works should be further discussed. For example, some prior works, such as CORRO, explores the problem of out-of-distribution context in COMRL.

**Experimental Designs Or Analyses:**

I have check the experiments in detail. Here are some issues:
- In Figure 3 and 4, I guess horizontal axes should denote training steps, not test steps, since the number goes to 1e5. Is this a typo? If so, why do returns decrease in Point-Robot and Ant-Goal in Figure 4?
- In the zero-shot setting, the returns of which episodes are reported?
- How are the behavior policies trained and the offline datasets collected for these experiments? How is the data quality, compared with the performance of the trained OMRL policies?

**Methods And Evaluation Criteria:**

The proposed method makes sense for the identified problems. The experimental settings properly follow prior works.

**Other Comments Or Suggestions:**

Please see issues above.

**Other Strengths And Weaknesses:**

Strengths:
- The paper unifies the problem of OOD context and task ambiguity into the problem of task uncertainty, which is important in the literature of COMRL. The proposed method makes sense.
- The proposed method can seamlessly integrate various approaches, including classification-based methods, reconstruction-based methods, and contrastive learning methods.

Weaknesses:
- As I mentioned above, some experimental details are confusing. Related works about OOD context and task uncertainty can be further discussed.

**Questions For Authors:**

I hope the authors address the issues in the writing of experiments and some related works.

**Relation To Broader Scientific Literature:**

The notion of context uncertainty has been well explored in online meta-RL, such as VariBAD [1]. But in OMRL, it is less explored. The proposed method is an intuitive and good solution to this problem in OMRL and I think it is a good contribution to the community.

[1] Varibad: A very good method for bayes-adaptive deep rl via meta-learning. 2019.

**Theoretical Claims:**

The paper makes no theoretical claims.

---

> ### Author Rebuttal · Authors · 2025-03-31
>
> We sincerely appreciate the reviewers’ constructive feedback. Below, we respond to each concern point by point. We will incorporate all the reviewers’ suggestions in the final version of the paper.
>
> > Reviewer:
> > - (line 40, right) ... However, CORRO [1] focuses on addressing OOD contexts in OMRL.
> > - Definitions 3.1, 3.2, 3.3 are more like an intuitive explanation and the definitions are either informal or not used in the following writing.
>
> Response: CORRO primarily focuses on improving representation robustness through data augmentation. It uses a CVAE-based approach to augment the dataset, enhancing the encoder’s robustness in OOD scenarios. However, its data augmentation relies on (s, a) pairs that have appeared in the dataset, inherently limiting its OOD generalizability. In contrast, our method identifies and evaluates ambiguous and potentially OOD contexts for adaptation purposes and is designed as a plug-in approach. Thus, we consider CORRO and our method orthogonal, meaning they can be combined to further improve robustness in OOD tasks.
>
> Regarding Definitions 3.1, 3.2, and 3.3, they are introduced to formally define key concepts in the paper.
>
> > Reviewer:
> > I have check the experiments in detail. Here are some issues:
> > - In Figure 3 and 4, ...
> > - In the zero-shot setting, the returns of which episodes are reported?
> > - How are the behavior policies trained and the offline datasets collected for these experiments? How is the data quality, compared with the performance of the trained OMRL policies?
>
> Response:
> Yes, the x-axis in Figures 3 and 4 should be “training step.”
> In the Zero-shot setting, we collect a trajectory using the context collection policy $\pi_\theta(s, z_0)$, and its return is reported as the Zero-shot performance. To further investigate the decline in baseline performance observed in Figure 4a during training, we visualized zero-shot trajectories (episode line 1) at different training steps at at https://anonymous.4open.science/r/CERTAIN-6073/experiment7.md, and found that as training progresses, the zero-shot trajectories gradually become more conservative, leading to a performance drop.
> Regarding the dataset, we strictly follow the data collection procedure used in baselines such as FOCAL. Specifically, in each task, we train an agent from scratch using SAC and collect data at training steps. Additionally, we have provided results on the performance of different methods under varying dataset qualities in <https://anonymous.4open.science/r/CERTAIN-6073/experiment2.md>. We found that dataset quality significantly affects all methods, particularly FOCAL. We hypothesize that FOCAL is more susceptible to spurious correlations compared to other methods.
>
> > Reviewer: I hope the authors address the issues in the writing of experiments and some related works.
>
> Response: We have conducted experiments on OOD contexts, where the context used for inferring task representations consists of 25% in-distribution samples (upper semicircle) and 75% out-of-distribution samples (lower semicircle) in <https://anonymous.4open.science/r/CERTAIN-6073/experiment1.md> and OOD tasks, where the goal is at  lower semicircle in <https://anonymous.4open.science/r/CERTAIN-6073/experiment4.md>.
> In the final version of the paper, we will include a more detailed discussion on OOD.

---

### Official Review · Reviewer_NvXZ · 2025-03-21

**Overall Recommendation:** 2

**Summary:**

This paper deals with the context ambiguity problem in context-based offline meta-reinforcement learning. The authors propose an uncertainty-aware context-collection algorithm to produce in-distribution, unambiguous contexts using heteroscedastic uncertainty estimates as rewards. Experiments are conducted in several MuJoCo environments.

## Update after rebuttal

This paper introduces an interesting idea into the field of COMRL and has the potential to make significant contributions, but probably needs more work regarding theoretical justifications and empirical validations. The authors are encouraged to revise the paper in these aspects.

**Claims And Evidence:**

Definitions 3.1-3.3 are not very clear. How are the conditional probabilities $p(c \mid \mathcal{M}\_i)$, $p(c \mid \mathcal{M}, \pi\_\beta)$ defined? Does $p(c \mid \mathcal{M}, \pi\_\beta)=0$ need to hold for a single $\mathcal{M}\_i$ or all of the tasks in the offline training dataset? What is the relationship between $\pi_\beta$ and the offline datasets $\mathcal{D}$? How is $\sigma$ defined and computed?
Furthermore, the empirical evidence is not very convincing (see the method and experiment parts below).

**Essential References Not Discussed:**

The paper misses comparison and discussion about a relevant prior work BOReL [1], which is cited but not thoroughly analyzed. BOReL seeks to address the ambiguous context problem by learning a meta-policy conditioned on the distribution of the latent posterior, making it also uncertainty-aware as CERTAIN. The idea of representing uncertainty as a latent distribution is common among related works, e.g. VariBAD [2] and PEARL [3], which are also cited but not discussed. Since CERTAIN takes another approach and estimates the uncertainty with the deterministic latent embedding, the paper could benefit greatly from more detailed comparisons and discussions.

[1] Dorfman, R., Shenfeld, I. and Tamar, A., 2021. Offline Meta Reinforcement Learning--Identifiability Challenges and Effective Data Collection Strategies. Advances in Neural Information Processing Systems, 34, pp.4607-4618.

[2] Zintgraf, L., Shiarlis, K., Igl, M., Schulze, S., Gal, Y., Hofmann, K. and Whiteson, S., VariBAD: A Very Good Method for Bayes-Adaptive Deep RL via Meta-Learning. In International Conference on Learning Representations.

[3] Rakelly, K., Zhou, A., Finn, C., Levine, S. and Quillen, D., 2019, May. Efficient off-policy meta-reinforcement learning via probabilistic context variables. In International conference on machine learning (pp. 5331-5340). PMLR.

**Experimental Designs Or Analyses:**

1. The empirical performance improvement of CERTAIN seems a bit marginal. In Fig. 3, CERTAIN variants only display a relatively clear advantage in Point-Robot and Walker-Rand-Params while being outperformed in Point-Robot and Ant-Goal in Fig. 4. The authors' explanations for this fail to convince. Also, the performance of baselines seems to decrease over time in Fig. 4(a) which is confusing.

2. What is the context collection policy in the first episode for the baseline methods? What is the x-axis "Test Steps" in Fig. 3 and 4?

3. (Minor) Both figures are a bit hard to read; it may help with readability to distinguish between baselines and CERTAIN+baselines using e.g. solid/dotted lines of the same color.

**Methods And Evaluation Criteria:**

While the use of heteroscedastic uncertainty estimates is common among existing works, they are primarily used to estimate aleatoric uncertainty (e.g., in [1]). In this paper, they are also used to estimate epistemic uncertainty for OOD cases. The uncertainty estimator $h\_\psi$ may struggle to produce correct estimates for OOD inputs, in which case the epistemic uncertainty estimation approach of [1] could be better suited. As the authors make specific claims about OOD contexts, more evidence in this scenario is needed to support these claims.

Furthermore, CERTAIN chooses to use a deterministic context encoder with an additional network $h\_\psi$ for uncertainty estimation, while a more natural choice could be to directly use a probabilistic context encoder for capturing this uncertainty (see the reference discussion below).

[1] Kendall, A. and Gal, Y., 2017. What uncertainties do we need in bayesian deep learning for computer vision?. Advances in neural information processing systems, 30.

**Other Comments Or Suggestions:**

Typo: Definition 3.1 missing subscript in $p(c \mid \mathcal{M}\_i)$; table captions mention the point-robot environment but include results from multiple environments (e.g. Tab. 1); Eq. (12) could be missing something, e.g. discount factor and max operators.

**Other Strengths And Weaknesses:**

The paper is well-motivated and the method makes intuitive sense. Fig. 1 and 2 are good illustrations and provide great clarity about the method. Visualizations are provided.

**Questions For Authors:**

1. Can CERTAIN reliably identify OOD contexts?

2. How does CERTAIN compare with Bayes Adaptive methods, e.g. BOReL?

**Relation To Broader Scientific Literature:**

This paper proposes a method to promote exploration and reduce task uncertainty in COMRL settings, which is a well-explored topic in the field of meta-RL. See the next section for specific discussions about the relevant literature.

**Theoretical Claims:**

N/A as there are no theorems or proofs.

---

> ### Author Rebuttal · Authors · 2025-03-31
>
> We sincerely appreciate the reviewers’ constructive feedback. Below, we respond to each concern point by point. We will incorporate all the reviewers’ suggestions in the final version of the paper.
>
> > Reviewer: Definitions 3.1-3.3 are not very clear ... How is $\sigma$ defined and computed?
>
> Response: $p(c|M_i)$ is defined as the probability of context $c$ occurring under the $i_{th}$ MDP $M_i$. The equation $p(c|M,\pi_\beta)=0$ holds for all tasks in the offline dataset. We refer to all collection strategies used during the dataset collection process as the behavior policy $\pi_\beta$. It can be a mixture of multiple policies, such as human experts, other agents, etc. We define the degree of context uncertainty as $\sigma$, which is a value greater than 0, calculated by a heteroscedastic network with the loss $\frac{L}{\sigma^2}+\log \sigma$. It should be noted that $\sigma$ captured the loss $L$ for the training dataset and, therefore, not only represents the heteroscedastic uncertainty but also the confidence of a specific encoder model (a.k.a the epistemic uncertainty).
>
> > Reviewer: The uncertainty estimator $h_\psi$ may struggle to produce correct estimates for OOD inputs ... more evidence in this scenario is needed to support these claims.
>
> Response: Yes, we acknowledge that $h_\psi$ does struggle to produce correct estimates for OOD inputs in theory. However, in many cases, it generalizes well to OOD context. We have added experiments in the point robot environment under OOD context conditions, where the context used for inferring task representations consists of 25% in-distribution samples (upper semicircle) and 75%  OOD samples (lower semicircle) at <https://anonymous.4open.science/r/CERTAIN-6073/experiment1.md>.
>
> > Reviewer: Furthermore, CERTAIN chooses to use a deterministic context encoder with an additional network for uncertainty estimation, while a more natural choice could be to directly use a probabilistic context encoder for capturing this uncertainty.
>
> Response: A probabilistic context encoder can indeed capture uncertainty within the training dataset, but it is typically restricted to VAE-based methods. In contrast, heteroscedastic uncertainty captures the magnitude of the loss, making it adaptable to classification and contrastive learning. This allows for a more generalizable representation of context uncertainty.
>
> > Reviewer:
> > 1. The empirical performance improvement of CERTAIN seems a bit marginal. In Fig. 3 ...
> > 2. What is the context collection policy in the first episode ...
> > 3. (Minor) Both figures are a bit hard to read ...
>
> Response:
> 1. Since our method does not directly improve the policy but provides a mechanism for collecting and weighting contexts during adaptation, it is reasonable that performance does not significantly improve when the uncertainty in the contexts is already low. To clarify, we have provided performance improvement percentages in tabular <https://anonymous.4open.science/r/CERTAIN-6073/experiment6.md>. The One-shot experiment in Figure 3 and the Zero-shot experiment in Figure 4 are not directly related in terms of performance. Instead, this result suggests that the quality of the collected context is not necessarily correlated with the return of consequent episodes. Therefore, it is crucial to prioritize high-quality contexts with lower uncertainty. To further investigate the decline in baseline performance observed in Figure 4a during training, we visualized zero-shot trajectories (episode line 1) at different training steps at at <https://anonymous.4open.science/r/CERTAIN-6073/experiment7.md>, and found that as training progresses, the zero-shot trajectories gradually become more conservative, leading to a performance drop.
>
> 2. The first trajectory for all methods is collected using $\pi_\theta(s, z_0)$. We sincerely appreciate the reviewer pointing out the incorrect labeling of the “X-axis”—it should indeed be “training step.”
> 3. Thank you for the reviewer’s suggestions regarding the figures. We have redrawn the experimental plots in <https://anonymous.4open.science/r/CERTAIN-6073/experiment5.md>.
>
> > Reviewer: The paper misses comparison and discussion about a relevant prior work BOReL ...
>
> Response: BOReL defines MDP ambiguity, which is different from our concept of context ambiguity. BOReL ultimately provides guidance on how to collect training data and how to use an oracle model to relabel existing datasets for correction. However, CERTAIN aims to identify uncertain contexts and improve the few-shot (one-shot) adaptation performance. Moreover, our method is a plug-in approach that can be applied to any context encoders.

---

> > ### Comment · Reviewer_NvXZ · 2025-04-03
> >
> > I thank the authors for their efforts in the rebuttal. Some of my remaining concerns are detailed below.
> >
> > > Clarity of definitions.
> >
> > It is still unclear what "the probability of context $c$ occurring under the $i$th MDP $\mathcal{M}\_i$" means. This is not a well-defined value without specifying a policy, does $p(c \mid \mathcal{M}\_i)>0$ mean e.g. $p(c \mid \mathcal{M}\_i, \pi\_\beta)>0$ or $\exists \pi, p(c \mid \mathcal{M}\_i, \pi)>0$? Definitions should be written as clearly as possible to avoid confusion. For example, it would also be better to explicitly write something like $p(c \mid \mathcal{M}, \pi):=\mathcal{P}\_0(s\_1)\prod\_{j=1}^K \pi(a\_j \mid s\_j) \mathcal{P}(s'\_j \mid s\_j, a\_j) \mathcal{R}(r\_j \mid s\_j, a\_j)$ (which I presume is what the authors intend to say) instead of language descriptions.
> >
> > > Epistemic uncertainty and OOD experiments.
> >
> > While I appreciate the additional figures, the theoretical and empirical justifications are still not sufficiently satisfactory. First of all, Eq. (10) does **not** estimate epistemic uncertainty; it is explicitly stated in (Kendall & Gal, 2017) that such a loss finds a single value for the model parameters and does not capture epistemic uncertainty over model parameters (see Sec. 2.2 under Eq. 5 in Kendall & Gal, 2017). A thorough review of (Kendall & Gal, 2017) and relevant materials is recommended for a deeper understanding of the differences between aleatoric and epistemic uncertainties. Furthermore, the additional OOD experiments are only presented as several examples in a toy environment and not very convincing. It's also unclear how the uncertainty estimates are obtained for the baseline methods, which seem to differ widely from those of CERTAIN (e.g. the first episode in FOCAL and the second episode of Classifier have very different uncertainty estimates from the CERTAIN variants).
> >
> > > Probabilistic encoder and BOReL.
> >
> > I fail to see why probabilistic context encoders can't be applied to other types of losses; this should be done fairly easily through the same reparameterization technique used for the reconstruction objective. The problem formulation and ultimate goal of BOReL are both similar to those of CERTAIN, for example, the concept of context uncertainty is captured by belief in BOReL (and VariBAD) while Definition 3.1 in CERTAIN is similar to Definition 3 (overlapping state-action) of BOReL. The authors are encouraged to compare with the core off-policy algorithm of BOReL, potentially under the same setting, e.g. with policy replaying and reward relabelling ablated.
> >
> > To conclude, this paper introduces an interesting idea into the field of COMRL and has the potential to make significant contributions, but probably needs more work, especially regarding uncertainty estimates and related works. The authors are encouraged to revise the paper in these aspects.

---

> > > ### Author Response · Authors · 2025-04-03
> > >
> > > We sincerely appreciate the reviewers’ further responses. Below, we address the newly raised concerns one by one:
> > >
> > > 1. We apologize for the confusion caused by the definition of $P(c|M_i)$. As the reviewer correctly inferred, the strict definition of $P(c|M_i)$ is based on all possible policies $\pi$, i.e., $P(c|M_i)=\int p(c|M_i, \pi) d_\pi> 0$. We will update the final version of the paper to reflect this stricter definition.
> > >
> > > 2. The uncertainty estimation $\sigma$ in our paper is only formally similar to “heteroscedastic uncertainty,” but its meaning is different. In (Kendall & Gal, 2017), heteroscedastic uncertainty is used to model Gaussian noise in the data, so the heteroscedastic loss must take the form of a Gaussian distribution. However, in our method, the original loss $L$ in Equation (10) is computed separately and is not limited to the form of a regression loss. Equation (10) is only used for learning $\sigma$. Given $L$, the theoretical minimum of Equation (10) is achieved when $\sigma = \sqrt{2L}$, allowing $\sigma$ to capture the magnitude of $L$ rather than merely modeling data noise.
> > >
> > > 3. The variance output by the probabilistic encoder models the distribution of the latent variable $z$, which we believe represents pure aleatoric uncertainty. In BOReL, the primary focus is on the issue of spurious correlations—specifically, if $(s, a)$ does not overlap, the encoder is likely to infer an incorrect MDP based on $(s, a)$, necessitating the relabeling of $(s, a)$. However, our paper does not focus on this issue. Instead, we are more concerned with cases where the context itself is ambiguous or OOD and how to mitigate its impact on task inference.

---

### Decision · Program_Chairs · 2025-05-01

**Decision:**

Accept (poster)

**Comment:**

This paper proposes CERTAIN, an uncertainty-aware context collection method for one-shot adaptation in context-based offline meta reinforcement learning. Leveraging heteroscedastic uncertainty estimates, CERTAIN can be used with prior COMRL methods to learn an uncertainty-aware task representation. The estimated uncertainty can further be used as penalties in context collection policy learning. Experiments are conducted in several commonly used MuJoCo benchmark environments and show improved performance.
The reviews are generally mixed with initial scores of [2, 2, 3, 3] and post-rebuttal [2, 3, 3, 3]. Overall, the reviewers appreciated the generality of CERTAIN, which can be integrated with various existing COMRL methods to improve performance, as demonstrated by the experiments. The problem of OOD context identification and mitigation is important to the broader COMRL community and real-world applications, for which CERTAIN provides a simple method. Existing concerns mainly involve the lack of theoretical analysis of CERTAIN, which the authors acknowledged during rebuttal. Empirical evidence in OOD scenarios is also weak, potentially due to the tight rebuttal timeline.

Overall, this is a borderline paper, with the benefits (generality of method, importance of problem) outweighing the downsides (theoretical foundations, OOD evaluation). So, I recommend a weak accept while encouraging the authors to improve the paper accordingly in the final version.